# Improving Generalization in Federated Learning with Model-Data Mutual Information Regularization: A Posterior Inference Approach

**Hao Zhang, Chenglin Li,* Nuowen Kan, Ziyang Zheng, Wenrui Dai, Junni Zou, Hongkai Xiong**
School of Electronic Information and Electrical Engineering
Shanghai Jiao Tong University

## Abstract

Most of existing federated learning (FL) formulation is treated as a point-estimate of models, inherently prone to overfitting on scarce client-side data with overconfident decisions. Though Bayesian inference can alleviate this issue, a direct posterior inference at clients may result in biased local posterior estimates due to data heterogeneity, leading to a sub-optimal global posterior. From an information-theoretic perspective, we propose FedMDMI, a federated posterior inference framework based on model-data mutual information (MI). Specifically, a global model-data MI term is introduced as regularization to enforce the global model to learn essential information from the heterogeneous local data, alleviating the bias caused by data heterogeneity and hence enhancing generalization. To make this global MI tractable, we decompose it into local MI terms at the clients, converting the global objective with MI regularization into several locally optimizable objectives based on local data. For these local objectives, we further show that the optimal local posterior is a Gibbs posterior, which can be efficiently sampled with stochastic gradient Langevin dynamics methods. Finally, at the server, we approximate sampling from the global Gibbs posterior by simply averaging samples from the local posteriors. Theoretical analysis provides a generalization bound for FL w.r.t. the model-data MI, which, at different levels of regularization, represents a federated version of the bias-variance trade-off. Experimental results demonstrate a better generalization behavior with better calibrated uncertainty estimates of FedMDMI.

## 1 Introduction

Federated learning (FL), as an emerging distributed learning framework, has garnered considerable attention [McMahan et al., 2017]. In this paradigm, clients collaborate to train a single global model, under the central coordination of a server. Notably, such a collaborative training proceeds without the necessity of sharing or exchanging the raw data of clients, thus providing a basic level of privacy protection. However, two primary challenges arise, specifically revolving around the intra-client data scarcity and inter-client data heterogeneity, since client models locally trained on these scarce and heterogeneous data are prone to overfitting and bias, leading to a diminished generalization performance for the globally aggregated model at the server.

A multitude of efforts have been made to address this issue from various perspectives, including federated optimization [Acar et al., 2021, Karimireddy et al., 2020], federated domain generalization [Nguyen et al., 2022, de Luca et al., 2022], federated knowledge distillation [Zhu et al., 2021, Afonin and Karimireddy, 2021], etc. Despite their progress, a majority of the existing FL formulations consistently treat it as a point-estimate (i.e., a single value estimated for each model weight) based

---

*Correspondence to: Chenglin Li <LCL1985@sjtu.edu.cn>.

38th Conference on Neural Information Processing Systems (NeurIPS 2024).

on the loss function optimized on local client data. From a probability theory perspective, however, using these single point-estimates as weights inherently poses a risk of overfitting on the scarce client-side training data. More crucially, this may also lead to overly confident decisions, due to the failure to provide reliable assessment of the uncertainty for models [Shridhar et al., 2019], which is indispensable in some safety-critical applications of federated learning, e.g., autonomous driving, healthcare, and finance.

Instead of single point-estimates, an alternative approach is to apply posterior inference over the model weights [Jospin et al., 2022], which is theoretically attractive for preventing overfitting to scarce training data and providing a natural way to assess uncertainty in weight estimates that can be further propagated into the model's prediction. Under the FL settings, however, little work has explored inferring a global posterior from the heterogeneous data across clients. To achieve the global posterior inference in FL, a typical framework comprises: *i)* each client initializing a local model based on global posterior parameters and independently conducting a local posterior inference; *ii)* the server receiving local posteriors and multiplicatively aggregating them based on global posterior decomposition. These two steps will then iterate until the global posterior converges. Following this framework, FedPA [Al-Shedivat et al., 2020] approximates these local posteriors via the Markov chain Monte Carlo (MCMC) method, and further reduces the computation cost. However, the data heterogeneity among clients may still lead to a biased local posterior inference, which in turn results in a sub-optimal global posterior. To mitigate this bias, FedEP [Guo et al., 2023] approximates the global posterior using an expectation propagation method, which, however, incurs extra storage and communication overhead. As a stateful method, FedEP is also not a suitable solution for low-participation FL scenarios. Consequently, the issue of biased local posteriors incurred by the data heterogeneity in federated posterior inference remains unresolved.

In this paper, from an information-theoretic perspective, we propose to infer the global posterior in FL by incorporating a global mutual information (MI) regularization between the model and data (FedMDMI), which has been proved to be an effective measure of generalization capability of the learning algorithms [Xu and Raginsky, 2017]. Under the FL settings, we show that the proposed global MI regularization can effectively alleviate the bias of local posterior incurred by data heterogeneity, and thus improve generalization capability of the global posterior. However, due to the non-exchange restriction of raw data in FL, it is impractical for the server to measure this global MI explicitly. We therefore turn to decomposing the global MI into local MI terms, converting the global objective with MI regularization into several locally optimizable objectives based on the client data. For these local objectives, we show that the optimal local posterior is a Gibbs posterior, a conclusion well-established in the field of PAC-Bayesian learning [Alquier, 2021]. Then, we employ the stochastic gradient Langevin dynamics (SGLD) method [Welling and Teh, 2011] to sample from this local Gibbs posterior, which provides an unbiased and efficient sampling-based posterior approximation. Finally, at the server, we approximate the sampling from the global Gibbs posterior by simply taking the average of samples from the local posteriors. This aggregation method has been successfully employed and proved to have a non-asymptotic convergence guarantee with the true global posterior in decentralized and federated scenarios [Gürbüzbalaban et al., 2021, Plassier et al., 2023]. Through in-depth analysis, we provide a theoretical guarantee of our FedMDMI by establishing a generalization bound for FL w.r.t. the model-data MI. As a byproduct, differential privacy protection can also be brought by our FedMDMI. Our main contributions are as follows.

- We introduce an information-theoretic approach for the global posterior inference in FL. It incorporates a global model-data MI term as regularization, which enhances generalization by alleviating bias of inferred local posteriors, and offers certain client-level privacy protection as byproduct.

- We establish a generalization bound for FL w.r.t. the global model-data MI, showing that regularizing this global MI leads to a reduction in the generalization error. At different levels of regularization, it also represents a federated version of the bias-variance trade-off.

- Extensive empirical results validate that FedMDMI outperforms the other point-estimate and Bayesian inference-based baselines, while providing well-calibrated uncertainty estimates.

## 2 Related Work

**Model-Data Mutual Information (MI).** The mutual information (MI) between model and data quantifies the information that a model contains about the raw data, and serves as an effective measure of the model's complexity. Extensive research [Xu and Raginsky, 2017, Asadi et al., 2018] has

revealed a theoretical connection between this MI and the generalization ability of learning algorithms in centralized learning. Russo and Zou [2016] further demonstrate that this MI can effectively bound and reduce the bias in data analysis for centralized learning. For distributed and federated learning, prior works [Yagli et al., 2020, Barnes et al., 2022, Sefidgaran et al., 2022] also establish generalization bounds based on local MI at clients. Recent studies [Chor et al., 2023, Sefidgaran et al., 2023] further explore the relationship between the generalization bound and communication rounds through the local MI. In contrast to these methods, our generalization bounds are based on global MI at the server, resulting in a tighter generalization bound. More significantly, we also provide a posterior inference method to estimate this MI.

**Bayesian Inference.** Compared to point-estimates of models, the posterior estimate is more suitable for FL with scarce local client data. Bayesian model ensemble learning-based methods, such as FedBE [Chen and Chao, 2020] and FedPPD [Bhatt et al., 2022], construct a robust global posterior distribution via ensemble learning and knowledge distillation, which, however, require additional auxiliary datasets at the server. Global posterior decomposition-based methods, on the other hand, decompose the global posterior into a product of client posteriors. Among them, FedPA [Al-Shedivat et al., 2020] proposes a computation- and communication-efficient framework for global posterior decomposition, where data heterogeneity may still result in a biased local inference and sub-optimal global posterior. While FedEP [Guo et al., 2023] introduces expectation propagation at the client side to obtain a sound global posterior at the server, this approach incurs additional storage and communication costs and may not be suitable for federated scenarios with low client participation. The work most closely related to ours is FALD [Plassier et al., 2023], which can be viewed as a special case of our FedMDMI when the hyperparameter $\alpha$ is set to 1. We will further demonstrate that this hyperparameter $\alpha$ is instrumental in controlling the tradeoff between fitting and generalization. Different from FALD, our FedMDMI further incorporates the global model as a more robust prior, and we derive a valuable generalization bound w.r.t. the model-data MI.

## 3 Preliminaries and Problem Statement

The objective of standard FL is to learn a single global model $w$ from $m$ clients via the following optimization problem:

$$\min_{w \in \mathbb{R}^d} L_{\mathcal{D}}(w) = \frac{1}{m} \sum_{i \in \mathcal{M}} L_{p(S_i)}(w), \quad (1)$$

Table 1: Summary of notations.

| | |
|---|---|
| $S_i, S$ | random variables denoting local, global data |
| $w$ | random variable denoting learned global model |
| $T, t$ | number, index of communication rounds |
| $K, k$ | number, index of local update step |
| $w_{t,k}^i$ | sample of client $i$'s model at round $t$ and step $k$ |
| $w_t$ | sample of aggregated server model at round $t$ |

where $L_{p(S_i)}(w) = \mathbb{E}_{z_i \sim p(S_i)}[\ell_i(w, z_i)]$ represents the local expected risk of the $i$-th client associated with data distribution $p(S_i)$ on local data $S_i$, and the loss function $\ell_i(w, z_i)$ is usually chosen as a negative log likelihood, i.e., $-\log p(z_i|w)$. We further define a set of random variables $S = \{S_1, \ldots, S_m\}$ as the global dataset, consisting of $m$ heterogeneous client datasets. Moreover, we let $\{z_i^j\}_{j=1}^{n_i}$ denote the local training dataset of size $n_i$ drawn independently from distribution $p(S_i)$. This setup also gives rise to the global empirical risk as:

$$\min_{w \in \mathbb{R}^d} \hat{L}_{\mathcal{S}}(w) = \frac{1}{m} \sum_{i \in \mathcal{M}} L_{S_i}(w), \quad (2)$$

where $L_{S_i}(w) = \frac{1}{n_i} \sum_{j=1}^{n_i} \ell_i(w, z_i^j)$ is the local empirical risk.

From a probability theory perspective, employing a single point-estimate as the model weight, as shown in Eq. (2), may render the model susceptible to overfitting, particularly when dealing with small and scarce datasets at clients. In addition, this point-estimate also makes the model incapable of correctly evaluating the uncertainty in the client's local training data, leading to overly confident decisions. Motivated by the Bayesian inference, we thus treat the global model $w$ as a random variable rather than a single point, and then estimate the global posterior of model $w$ given the global dataset $S$. Subsequently, the objective of FL converts to minimizing the posterior expected loss:

$$\min_{p(w|S)} L_{\mathcal{S}}(w) = \mathbb{E}_{p(w|S)} \left[ \frac{1}{m} \sum_{i \in \mathcal{M}} L_{S_i}(w) \right], \quad (3)$$

where $p(w|S) = p(w|S_1, \ldots, S_m)$ denotes the global posterior. By taking an additional expectation over $p(w|S)$, we are then able to evaluate the risk of learned global posterior rather than a single value-estimate of model weights $w$.

However, it is intractable to directly infer the global posterior, since raw data of the clients cannot be exchanged in FL. A widely adopted solution is the **global posterior decomposition** [Neiswanger et al., 2014, Al-Shedivat et al., 2020], which decomposes the global posterior into a product of client's local posteriors, i.e., $p(w|S) \propto \tau \prod_{i \in \mathcal{M}} p(w|S_i)$, where $\tau \triangleq \frac{p(w)}{\prod_{i=1}^{m} p_i(w)}$ denotes the ratio of global prior to the product of client priors, which can be viewed as a constant based on prior assumptions, e.g., the Gaussian priors. See Appendix A.4.1 for the detailed derivation of this decomposition.

Based on the global posterior decomposition, the update process to optimize Eq. (3) can then be performed as follows: *i)* each client initializes the model based on global posterior received from the server and independently learns a local posterior (e.g., by variational inference or Markov Chain Monte Carlo); *ii)* the server collects the local posteriors from clients and multiplicatively aggregates them to obtain the global posterior, which is then sent back to the clients for the next round of update. This iterative process continues until the global posterior converges, as depicted in Eq. (4):

$$\text{Server aggregation: } p(w^*|S) = \prod_{i \in \mathcal{M}} p(w^*|S_i); \quad \text{Client update }^2 \colon p(w^*|S_i) = \underset{p(w|S_i)}{\arg\min} \mathbb{E}_{p(w|S_i)} L_{S_i}(w). \quad (4)$$

This iterative update process is similar to FedAvg [McMahan et al., 2017], but has distinctive features: *i)* each client estimates a distribution $p(w|S_i)$ instead of a single value of $w$, and *ii)* the global aggregation on the server is changed from weighted averaging to multiplication. FedPA [Al-Shedivat et al., 2020] has enabled this process and further reduced computation and communication costs through federated least squares.

Though this posterior inference helps alleviate overfitting in scenarios with scarce client data as compared to single point-estimates, the heterogeneity of the data remains an issue. The independently learned local posteriors are susceptible to shifts induced by data heterogeneity, hindering their ability to generalize to data from other clients. Subsequently, the aggregation of biased local posteriors may lead to a sub-optimal global posterior [Guo et al., 2023]. This then raises a fundamental question: how can we achieve a globally well-generalized posterior in federated heterogeneous scenarios.

## 4 Proposed FedMDMI

To reduce the model's dependency on heterogeneous data and mitigate biased local posteriors incurred by such heterogeneity, we are motivated to introduce an additional regularization on the mutual information (MI) between the model and data (MD). In this section, we elaborate on our FedMDMI with an overview illustrated in Figure 1, and procedure summarized in Algorithm 1 in Appendix A.1.

### 4.1 Compressing Information in Weights by Model-Data MI Constraint

The model-data mutual information is denoted as $I(w; S)$, which quantifies relevance between the model $w$ and input data $S$, and also serves as a measure of complexity of the learned model for generalization analysis. By incorporating this information-theoretic constraint as regularization, the original optimization formulation in Eq. (3) is converted to:

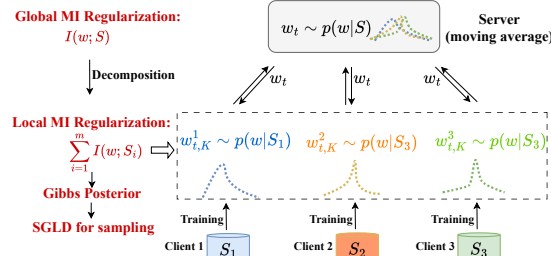

$$\min_{p(w|S)} \quad \mathbb{E}_{p(w|S)} \left[ \frac{1}{m} \sum_{i \in \mathcal{M}} L_{S_i}(w) \right], \quad (5)$$
$$\text{s.t.} \quad I(w; S) < I_c,$$

Figure 1: Overview of FedMDMI: under MI constraint, each client uploads the sample from the local posterior to server and subsequently downloads the aggregated global sample from the global posterior.

where the introduced MI constraint compresses information stored in model weights, and $I_c$ denotes the upper bound of the global MI. Namely, if less information is extracted by the model from the global heterogeneous dataset, it is less likely that overfitting occurs. Furthermore, this MI constraint is also helpful in mitigating the bias incurred by data heterogeneity. If we denote the bias factor as $\delta$ that affects the generation of local heterogeneous data (such as the difference

---

[2]Here, clients are not required to estimate an optimal local posterior before uploading. Similar to FedAvg [McMahan et al., 2017], the local posteriors are uploaded after a certain number of updates.

in clients' locations, preferences, or habits), we then have the Markov chain $\delta \to S \to w$, implying $I(w; \delta) \leq I(w; S)$. As a consequence, we can diminish $I(w; S)$ to constrain $I(w; \delta)$, thereby rendering the model $w$ insensitive to the bias factor $\delta$ from the diverse clients.

In essence, this newly introduced model-data MI term $I(w; S)$ constrains the complexity of model space for searching, and also encourages the model $w$ to learn essential information from the global heterogeneous data $S$. Note that various studies in FL have also attempted to induce the model to learn invariant information across clients through another crucial MI constraint, which is the MI between input data and output representation, commonly known as the information bottleneck [Shwartz-Ziv and Tishby, 2017] regularization. However, accessing this MI constraint usually requires sharing additional feature representations of the data across clients, which is infeasible in FL [Zhang et al., 2023]. In contrast, our proposed model-data MI term not only requires no additional information to be shared with clients, but also provides client-level privacy protection as a byproduct, which will be discussed in detail in Section 5.3.

## 4.2 Global Model-Data MI Decomposition

While our new formulation in Eq. (5) is conceptually promising, it is still infeasible to directly measure the global model-data MI term $I(w; S)$ in FL, where we are constrained to only leverage the distributed data at individual clients. Alternatively, we decompose it into several local MI terms $I(w; S_i)$, which can be determined locally by these clients.

**Proposition 4.1.** *(Global Model-Data MI Decomposition). Suppose that $S = \{S_1, \ldots, S_m\}$ consists of data from the $m$ clients and $S^{i-1} \triangleq \{S_1, \ldots, S_{i-1}\}$, then based on the chain rule of MI, we have:*

$$I(w; S) = \sum_{i=1}^{m} \left[ I(w; S_i) - I\left(S_i; S^{i-1}\right) \right] \leq \sum_{i=1}^{m} I(w; S_i). \tag{6}$$

*Proof.* See Appendix A.4.3 for the detailed proof. $\square$

Therefore, instead of $I(w; S)$, we can constrain its upper bound $\sum_{i=1}^{m} I(w; S_i)$ in Eq. (5). By further introducing a Lagrange multiplier $\alpha \geq 0$, Eq. (5) then re-formulates to:

$$\min_{p(w|S)} \mathbb{E}_{p(w|S)} \left[ \frac{1}{m} \left( \sum_{i \in \mathcal{M}} L_{S_i}(w) + \alpha I(w; S_i) \right) \right], \tag{7}$$

where $\alpha$ is also viewed to balance the fitting and generalization. Based on the global posterior decomposition, the iterative update process to optimize Eq. (7) becomes:

$$\text{Server aggregation: } p(w^*|S) = \prod_{i \in \mathcal{M}} p(w^*|S_i); \tag{8}$$

$$\text{Client update: } p(w^*|S_i) = \underset{p(w|S_i)}{\arg\min} \mathbb{E}_{p(w|S_i)} \left[ L_{S_i}(w) + \alpha I(w; S_i) \right]. \tag{9}$$

At each round, client update needs not to fully infer optimal local posterior. Consistent with FedAvg, it only updates for a certain number of iterations before sending learned local posterior to server.

## 4.3 Local Posterior Inference

For the client update in Eq. (9), the second term $I(w; S_i)$ can be expressed as:

$$I(w; S_i) = \mathbb{E}_{p(S_i)} \left[ \text{KL} \left[ p(w|S_i) || p_i(w) \right] \right], \tag{10}$$

which denotes the expectation of Kullback-Leibler (KL) divergence between the posterior $p(w|S_i)$ and marginal distribution $p_i(w)$[3] over the local data distribution $p(S_i)$. Notably, this KL divergence plays a key role in the well-known PAC (Probably Approximately Correct)-Bayes bound[4]. PAC-Bayes learning [Alquier, 2021] provides a tight generalization bound for learning algorithms, where the KL divergence between the posterior $p(w|S_i)$ and prior $p_i(w)$ is a dominant term within this bound. For a fixed prior, pioneer works [Xu and Raginsky, 2017, Alquier et al., 2016] have found an optimal posterior to minimize this bound, which is often referred to as the Gibbs posterior. Namely, the optimal posterior follows a typical Gibbs distribution.

---

[3] Specifically, $p_i(w) \triangleq \mathbb{E}_{p(S_i)}[p(w|S_i)]$, which is also called the oracle prior.

[4] The PAC-Bayes bound still holds even when $p_i(w)$ and the posterior $p(w|S_i)$ are chosen arbitrarily.

**Lemma 4.2.** *[Xu and Raginsky, 2017] The Gibbs posterior is the minimum of the objective for client update in Eq. (9):*

$$p(w|S_i) = \frac{1}{B_i} \exp\left[-\frac{1}{\alpha}\left(L_{S_i}(w) - \alpha \log p_i(w)\right)\right], \tag{11}$$

*where $B_i$ is a normalization factor.*

*Proof.* See Appendix A.4.4 for the detailed proof. $\square$

One possible way to obtain this Gibbs posterior is to seek its variational approximation (VA) [Alquier et al., 2016], which generally relies on simplified posterior distributions. However, a primary drawback of VA is its tendency to yield biased posterior estimates for complex posterior distributions. What is worse, in FL the data heterogeneity across clients itself contributes already to biased local posteriors, and this bias may be exacerbated by VA. Moreover, this method also results in at least doubling the communication overhead due to transmission of both the mean and covariance matrices. In contrast, Markov chain Monte Carlo (MCMC) methods offer an alternative class of sampling-based posterior approximations that are unbiased [Vadera et al., 2020], albeit with a slower convergence. Thus, we adopt the stochastic gradient Langevin dynamics (SGLD) [Welling and Teh, 2011], an MCMC method that has been proven effective and scalable in large-scale posterior inference problem.

Specifically, SGLD draws samples from the Gibbs posterior, by using the stochastic gradient update:

$$w_{t,k}^i = w_{t,k-1}^i - \eta_L^{t,k} \nabla U_i\left(w_{t,k-1}^i\right) + h_{t,k}, \tag{12}$$

where $w_{t,k}^i$ represents client $i$'s model at round $t$ and step $k$, $\eta_L^{t,k}$ is the local step size, $h_{t,k}$ is a noise variable sampled from $\mathcal{N}\left(\mathbf{0}, \sqrt{2\eta_L^{t,k}\alpha}\mathbf{I}\right)$ with $\mathbf{I}$ being the identity matrix, and

$$\nabla U_i\left(w_{t,k-1}^i\right) = \nabla\left(L_{S_i}(w_{t,k-1}^i) - \alpha \log p_i(w)\,|_{w=w_{t,k-1}^i}\right) \tag{13}$$

is an unbiased estimate of gradient. Note that $U_i$ corresponds to the exponent in the Gibbs posterior of Eq. (11), and $p_i(w)\,|_{w=w_{t,k-1}^i}$ represents the prior $p_i(w)$'s probability density value at $w = w_{t,k-1}^i$.

### 4.4 Global Posterior Aggregation and Prior Selection

In theory, if the step size is annealed as $\eta_L^{t,k} \to 0$, the client update sequence $w_{t,k}^i$ converges to the local Gibbs posterior in Eq. (11) with sufficiently large $k$ and $t$. The question then becomes: how can we obtain samples from the global posterior (i.e. the product of local Gibbs posteriors) expressed as

$$p(w|S) = \prod_{i\in\mathcal{M}} p(w|S_i) = \frac{1}{B'} \exp\left[-\frac{1}{\alpha}\sum_{i\in\mathcal{M}}\left(L_{S_i}(w) - \alpha \log p_i(w)\right)\right], \tag{14}$$

based on the samples drawn from these local posteriors.

Recent studies [Gürbüzbalaban et al., 2021, Plassier et al., 2023] have shown that when clients utilize SGLD for posterior inference in distributed or federated settings, leveraging the mean of samples from local posteriors to approximate samples from the target global posterior $p(w|S)$, there is a non-asymptotic convergence guarantee[5]. In other words, we can approximate samples from the global posterior by simply taking an average of the samples drawn from the local posteriors. Specifically, in the local posterior inference, SGLD introduces uncertainty into the predictive estimates by incorporating Gaussian noises, and samples the local model $w_{t,k}^i$ from the local posterior $p(w|S_i)$ in Eq. (11) through a Markov chain with steps:

$$\Delta w_{t,k}^i \sim \mathcal{N}\left(-\eta_L^{t,k}\nabla U_i\left(w_{t,k-1}^i\right), \sqrt{2\eta_L^{t,k}\alpha}\mathbf{I}\right), \tag{15}$$

where $\Delta w_{t,k}^i \triangleq w_{t,k}^i - w_{t,k-1}^i$. After the client performs $K$ steps and uploads the model change, the server can approximate a sample $w_{t+1}$ from the global posterior $p(w|S)$ in Eq. (14) simply by taking an averaging (i.e., $\Delta w_t \triangleq w_{t+1} - w_t = \frac{1}{m}\sum_{i\in\mathcal{M}}\Delta w_t^i$), through a global Markov chain with steps:

$$\Delta w_t \sim \mathcal{N}\left(-\frac{1}{m}\sum_{i\in\mathcal{M}}\sum_{k=1}^K \eta_L^{t,k}\nabla U_i\left(w_{t,k-1}^i\right), \frac{1}{m}\sum_{k=1}^K\sqrt{2\eta_L^{t,k}\alpha}\mathbf{I}\right). \tag{16}$$

---

[5]While this convergence bound $\mathcal{O}((1-\gamma\mu/8)^{KT}+1/m)$ is established only for $\mu$-strongly convex functions, empirical evaluations in Plassier et al. [2023] and our results show that it still holds in non-convex settings.

We then discuss how the prior is chosen in Eq. (13). This oracle prior (i.e., $p_i(w) \triangleq \mathbb{E}_{p(S_i)}[p(w|S_i)]$) renders the mutual information the tightest, which, however, is infeasible to obtain. In fact, we can use an arbitrary prior $r(w)$ (e.g., $\mathcal{N}(\mathbf{0}, \mathbf{I})$) to approximate $p_i(w)$, based on following upper bound:

$$I(w; S_i) = \mathbb{E}_{p(S_i)}\big[\mathrm{KL}\,\big(p(w|S_i)\|r(w)\big)\big] - \mathrm{KL}\,\big(p_i(w)\|r(w)\big) \le \mathbb{E}_{p(S_i)}\big[\mathrm{KL}\,\big(p(w|S_i)\|r(w)\big)\big]. \tag{17}$$

For the arbitrary prior $r(w)$ to achieve a smaller MI, it must essentially predict the posterior [Dziugaite et al., 2021]. Thus, we consider using the information of global model $w_t$ as the prior of clients for update of the next round. Based on Eq. (16) for updating $w_t$, we then define $p_i(w) \triangleq \mathcal{N}(w|\mu_{t-1}, \Sigma_{t-1})$, where

$$\mu_{t-1} = w_{t-1} - \frac{1}{m}\sum_{i\in\mathcal{M}}\sum_{k=1}^{K}\eta_L^{t-1,k}\nabla U_i\left(w_{t-1,k-1}^i\right), \; \Sigma_{t-1} = \frac{1}{m}\sum_{k=1}^{K}\sqrt{2\eta_L^{t-1,k}\alpha}\mathbf{I}. \tag{18}$$

Here the variance captures the uncertainty introduced by all the clients at the previous round $t-1$. Meanwhile, Zhang et al. [2022] also show that using the global model as a local prior can alleviate local overfitting. Additionally, $w_t$ is only a sample drawn from the global posterior. To better estimate the mean of this Gaussian prior, we approximate it with a global moving average, which is also utilized to accelerate the convergence[6], as shown in Line 16 of Algorithm 1 in Appendix A.1.

## 5 FedMDMI Analysis

### 5.1 Generalization Analysis via Model-Data MI

We first provide an information-theoretic generalization bound in terms of the model-data MI for FL. In FL settings, it is crucial to consider both the gaps arising from the unseen client data (i.e., participating error), and the gaps stemming from the unseen client distributions (i.e., participation gap). Following the framework proposed by Yuan et al. [2021] and Hu et al. [2023], we re-define the more general population risk in FL as:

$$L_\mathcal{P}(w) = \mathbb{E}_{p(S_i)\sim P}\left[\mathbb{E}_{z_i\sim p(S_i)}[\ell_i(w, z_i)]\right], \tag{19}$$

where we denote $P$ as a meta-distribution on $D$, and $D$ is the set of all distributions $p(S_i)$. This formulation takes into account the participation gap, as compared to Eq. (1). By recalling the global empirical risk $L_\mathcal{S}(w)$ in Eq. (3), we define the expected generalization error as $\mathbb{E}\left[L_\mathcal{P}(w) - L_\mathcal{S}(w)\right]$.

**Theorem 5.1.** *(Generalization Bounds for FL). Suppose that $\ell_i(w, z_i^j)$ for all $i \in \mathcal{M}$ is bounded by $C$ and independent, then the expected generalization error satisfies:*

$$\mathbb{E}\left[L_\mathcal{P}(w) - L_\mathcal{S}(w)\right] \le \sqrt{\frac{C^2 I(w; S)}{2mn}} + \sqrt{\frac{C^2 I(w; D)}{2m}}, \tag{20}$$

*where $m$ is the number of clients, $n$ is the number of samples on the client (assuming, without loss of generality, that the number of samples for all clients is equal), and $D$ is the set of distributions $p(S_i)$.*

*Proof.* See Appendix A.4.5 for the detailed proof. $\square$

*Remark* 5.2. On the RHS of Eq. (20), $I(w; S)$ relates to participating generalization error and serves as the regularized mutual information term that can be estimated in our FedMDMI, while $I(w; D)$ relates to the participation gap that cannot be estimated due to unavailability of the non-participating clients. Moreover, this generalization bound $\mathcal{O}\left(\frac{1}{\sqrt{mn}} + \frac{1}{\sqrt{m}}\right)$ matches with the current bound [Hu et al., 2023].

*Remark* 5.3. Several prior works [Yagli et al., 2020, Barnes et al., 2022, Sefidgaran et al., 2022] have also established a generalization bound based on clients' local MI $I(w; S_i)$ in distributed (or federated) learning. Additionally, some studies [Chor et al., 2023, Sefidgaran et al., 2023] in FL have explored generalization upper bounds w.r.t. the local time-varying mutual information $I(w_t, S_i^t)$. In contrast to them, our generalization bound is based on a participation gap framework, with the first term (global MI $I(w; S)$) exhibiting a tighter bound than the previous local MI $\sum_i I(w; S_i)$. For more details, we provide more elaboration of this distinction in Appendix A.4.2.

---

[6]We have also incorporated this technique into the other baselines in our experiments for a fair comparison.

## 5.2 Fitting vs. Generalization with Data Heterogeneity

As demonstrated in Theorem 5.1, the MI $I(w; S)$, acting as a measure of the effective complexity of a model, controls participating generalization capability. The hyperparameter $\alpha$ is employed in Eq. (7) to regulate this MI, thereby balancing fitting and generalization of the learned model. Here, we empirically investigate this trade-off under two non-iid settings using the CIFAR-10 dataset. The experiments involve 100 clients with a 10% participation rate (refer to Section 6 for details). The train and test errors of the global model are plotted in Figure 2 under varying $\alpha$.

It can be seen that with the increase of $\alpha$, the train error increases slowly, while the test error decreases initially and then increases. This suggests that models with lower complexity (larger $\alpha$) tend to underfit, while those with higher complexity (smaller $\alpha$) tend to over-fit. The MI $I(w; S)$ allows us to recover a federated version of the bias-variance trade-off. Moreover, as $\alpha$ increases, the gap between the test errors under different data heterogeneity decreases, indicating that the introduced $I(w; S)$ can render $w$ insensitive to data heterogeneity and alleviate the bias caused by such heterogeneity.

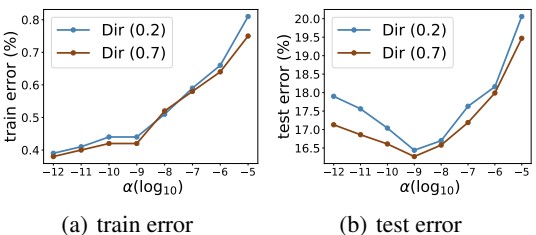

(a) train error      (b) test error

Figure 2: Train and test errors of the global model on CIFAR-10 dataset, where Dir (0.2) has a higher degree of data heterogeneity than Dir (0.7).

## 5.3 Client-Level Privacy Protection

The proposed FedMDMI can also provide client-level privacy protection, according to the analysis on MI and posterior inference based on SGLD sampling. First, the proposed MI regularizer $I(w; S)$ directly quantifies the extent to which the model memorizes data. Conversely, it reflects the degree of data information leakage. In the extreme case when $I(w; S) = 0$, $w$ becomes entirely independent of $S$ (e.g., akin to white noise) and leaks no information about the data. Thus, we restrict this MI to some extent for privacy protection. Second, SGLD in the local posterior inference has been widely demonstrated to provide strict differential privacy [Wang et al., 2015, Dziugaite and Roy, 2018]. Following these works and assuming that $\ell_i(w, z_i^j)$ is $L$-smooth, when $k \geq \mathcal{O}\left(\frac{\sqrt{\alpha}\epsilon^2}{\log(2/\delta)}\right)$, our FedMDMI also preserves a client-level $(\epsilon, \delta)$-differential privacy. See Appendix A.4.6 for details.

## 5.4 Limitation and Complexity Analysis

Though FedMDMI enjoys multi-fold benefits, including mitigating bias and overfitting on heterogeneous data to enhance generalization, and differential privacy as a byproduct, it may also have limitations. **First**, both the global posterior decomposition and global MI decomposition rely on the assumption that the global likelihood is conditionally independent given $w$. While this assumption is commonly adopted in embarrassingly parallel [Neiswanger et al., 2014] and federated scenarios [Guo et al., 2023, Al-Shedivat et al., 2020], it may not be applicable in certain extreme FL scenarios. **Second**, we do not delve into the convergence rate of FedMDMI under non-convex settings. While numerous studies [Gürbüzbalaban et al., 2021, Plassier et al., 2023] have analyzed the convergence of SGLD for strongly convex objective in distributed and federated posterior inference, and empirically demonstrated its effectiveness for non-convex objective, the theoretical convergence of SGLD remains an open question when applied to the non-convex objectives in FL.

**Last**, regarding the complexity analysis, we begin by defining the dimensions of the neural network as $d$. At each communication round, we adopt the SGLD to estimate the local posterior. Specifically, compared to the SGD employed in FedAvg and FedBE, our FedMDMI entails an additional step of generating the Gaussian noise with $d$ dimensions, which is subsequently incorporated into each model update iteration. This results in an additional $\mathcal{O}(d)$ time and $\mathcal{O}(d)$ memory. In contrast, at each client, FedPA uses dynamic programming to approximate the inverse matrix $d \times d$ of the neural network, introducing an additional $\mathcal{O}(l^2 d)$ time and $\mathcal{O}(ld)$ memory, where $l$ is the number of posterior samples. Similarly, FedEP(L) also requires approximating the covariance as the inverse Hessian, introducing an additional $\mathcal{O}(d^3)$ time and $\mathcal{O}(d^2)$ memory. For communication and aggregation at the server, our FedMDMI, along with FedAvg and FedPA, requires $\mathcal{O}(md)$ time and $\mathcal{O}(md)$ memory, where $m$ denotes the number of clients. In contrast, FedEP requires $\mathcal{O}(md)$ time and $\mathcal{O}(md^2)$ memory. Note

Table 2: Performance comparison (with 5 random seeds) under various settings, where a smaller Dirichlet parameter indicates a higher data heterogeneity, and L and H indicate low and high participation rates, respectively. Bold numbers indicate the best performance.

| Dataset | Setting | Top-1 Test Accuracy (%). | | | | | | | | |
|---|---|---|---|---|---|---|---|---|---|---|
| | | FedAvg | FedM | MimeLite | SCAFFOLD | FedBE | FedPA | FedEP (I) | FALD | FedMDMI (Ours) |
| CIFAR-10 | Dir (0.2)-L | 79.26±0.97 | 82.12±0.48 | 79.54±1.02 | 82.11±0.39 | 81.49±0.57 | 82.29±0.70 | 82.63±0.94 | 82.56±0.63 | **83.42**±0.39 |
| | Dir (0.7)-L | 80.01±1.29 | 82.46±0.90 | 80.25±0.82 | 82.25±0.47 | 82.06±0.68 | 82.61±0.56 | 83.18±0.54 | 83.06±0.67 | **83.81**±0.81 |
| | Dir (0.2)-H | 79.68±0.35 | 82.17±0.52 | 79.89±0.69 | 82.56±0.39 | 81.25±0.82 | 82.78±0.62 | 83.30±0.77 | 82.84±0.92 | **83.56**±0.44 |
| | Dir (0.7)-H | 80.31±0.60 | 82.69±1.01 | 80.12±0.48 | 83.04±0.62 | 82.33±0.38 | 82.93±0.70 | **83.79**±0.58 | 83.12±1.13 | 83.76±0.61 |
| CIFAR-100 | Dir (0.2)-L | 40.35±0.77 | 47.13±0.49 | 40.25±0.59 | 47.76±0.31 | 44.55±0.28 | 48.59±0.90 | 48.89±1.17 | 48.43±0.50 | **49.46**±0.55 |
| | Dir (0.7)-L | 41.29±0.59 | 47.89±0.98 | 41.99±0.56 | 48.14±0.91 | 45.76±0.47 | 49.45±0.67 | 49.55±0.85 | 48.86±0.60 | **50.45**±0.85 |
| | Dir (0.2)-H | 40.32±0.50 | 47.02±0.62 | 40.82±0.68 | 47.49±0.91 | 44.82±0.46 | 48.51±0.87 | 49.08±0.67 | 48.23±0.27 | **49.70**±0.61 |
| | Dir (0.7)-H | 42.35±0.90 | 48.67±0.62 | 42.39±0.51 | 47.99±0.62 | 46.29±0.45 | 49.66±0.47 | 50.02±0.49 | 49.55±0.83 | **50.71**±0.91 |
| Shakespeare | non-iid-L | 46.11±0.41 | 50.47±0.55 | 46.60±0.73 | 49.78±0.35 | - | 50.11±0.68 | 51.39±0.81 | 51.08±0.73 | **52.93**±0.59 |
| | non-iid-H | 46.89±0.49 | 51.10±0.52 | 47.19±0.78 | 48.99±0.70 | - | 51.26±0.64 | 52.06±0.58 | 51.12±0.57 | **53.37**±0.71 |

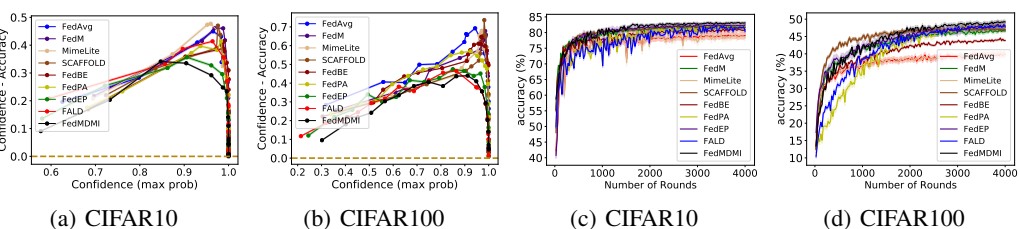

(a) CIFAR10          (b) CIFAR100          (c) CIFAR10          (d) CIFAR100

Figure 3: (a) and (b): Reliability diagrams for uncertainty estimates, where confidence is the value of the maximum softmax output. A perfectly calibrated model shows no difference between accuracy and confidence, which is represented by a dashed brown line. Points below this line indicate under-confident predictions, while points above this line correspond to overconfident predictions. (c) and (d): Test accuracy vs. number of communication rounds with Dir (0.2) and participation rate 5%, where curves are averaged over 5 random seeds.

that FedBE not only requires $\mathcal{O}((m+1+l)d)$ time and $\mathcal{O}((m+1+l)d)$ memory, where $l$ denotes the number of global model samples, but also performs the knowledge distillation at the server using unlabeled data, which is both memory- and time-intensive.

## 6   Experiments

**Datasets and Models.** We evaluate our FedMDMI on three benchmark datasets: CIFAR-10, CIFAR-100, and Shakespeare, with heterogeneous data splits. Specifically, for CIFAR-10 and CIFAR-100, each client has an uncertain number of classes through setting client sample labels according to the Dirichlet distribution. For example, with Dirichlet (0.2) each client has 80% samples which belong to mostly three or four different classes, while with Dirichlet (0.7) each client has 80% samples which belong to mostly five or six different classes. We use the CNN and RNN models similar to the prior works [McMahan et al., 2017, Acar et al., 2021]. Detailed experiment setup can be found in Appendix A.2.1.

**Comparison Methods.** Methods for comparison include the single point-estimate-based approaches: FedAvg [McMahan et al., 2017], FedM (FedAvg with momentum moving average) [Hsu et al., 2019], MimeLite [Karimireddy et al., 2021], and SCAFFOLD [Karimireddy et al., 2020], as well as the Bayesian inference-based approaches: FedBE [Chen and Chao, 2020], FedPA [Al-Shedivat et al., 2020], FedEP (I) [Guo et al., 2023] , and FALD [Plassier et al., 2023]. Note that Guo et al. [2023] and Plassier et al. [2023] also propose various variants of their methods. In our evaluation, we opt for FedEP (I) and FALD, as they demonstrate superior overall performance. We present additional results of the other baseline (i.e., $\beta$-PredBayes [Hasan et al., 2024]) in Appendix A.3.5. In addition, for a fair comparison, we also incorporate the momentum moving average in the global model aggregation step of FedM, FedPA, FedEP, and FALD.

**Implementation.** We evaluate the performance of the global model after 4000 communication rounds for CIFAR-10 and CIFAR-100, and after 1000 rounds for Shakespeare, which is trained with 100 clients. We set the high (H) and low (L) client participation rates as 10% and 5%, respectively. Clients are uniformly sampled at random without replacement at each round. The learning rate and hyperparameters for all approaches are individually tuned over a grid search. See Appendix A.2.3 for the additionally detailed settings of hyperparameters. Implementable code for evaluation of our FedMDMI is available at: `https://github.com/haozzh/FedMDMI`.

**Experiment Setting regarding Uncertainty Quantification for Image Tasks.** Uncertainty estimation is crucial for decision-making. Following Guo et al. [2017], Maddox et al. [2019], we utilize the expected calibration error (ECE) as a calibration metric of predictive uncertainty. To calculate ECE for a given model, we divide the test samples into 20 bins based on the model's confidence. Subsequently, we compute the absolute difference between the average confidence and accuracy within each bin and average these differences across all the bins. Additionally, reliability diagrams are plotted to illustrate the discrepancy between a method's confidence in its predictions and its actual accuracy. For a well-calibrated model, the discrepancy should be close to zero in each bin.

**Performance Evaluation.** The experimental results of all comparison methods under different non-iid settings are presented in Table 2 and Figures 3(c) and 3(d). In most cases, FedMDMI outperforms the other algorithms on the three datasets with varying heterogeneous data and client participation rates. This superior performance can be attributed to the proposed model-data MI regularization, which encourages the distributed clients to learn essential information and alleviates bias in the inferred local posteriors, thereby enhancing the generalization capability of the global model. It is worth noting that the convergence rate of the proposed FedMDMI may not be the fastest, especially when compared to the optimization-based method, SCAFFOLD. One potential explanation is because of our use of stochastic gradient Langevin dynamics (SGLD) to approximate the posterior, which often suffers from a slow convergence rate due to the variance introduced by the stochastic gradient [Wang et al., 2021a]. However, compared to SCAFFOLD, our method not only achieves a superior generalization performance, but also offers an improved calibration of uncertainty. Due to space limit, we present some additional experimental results and additional discussion in Appendix A.3.

Table 3: ECE (averaged over 5 random seeds).

| Method | CIFAR-10 | | CIFAR-100 | |
|---|---|---|---|---|
| | Dir (0.2)-L | Dir (0.7)-L | Dir (0.2)-L | Dir (0.7)-L |
| FedAvg | $0.169 \pm 0.0039$ | $0.165 \pm 0.0042$ | $0.429 \pm 0.0030$ | $0.432 \pm 0.0028$ |
| FedM | $0.159 \pm 0.0025$ | $0.169 \pm 0.0036$ | $0.468 \pm 0.0027$ | $0.459 \pm 0.0035$ |
| MimeLite | $0.182 \pm 0.0041$ | $0.178 \pm 0.0034$ | $0.461 \pm 0.0029$ | $0.470 \pm 0.0022$ |
| SCAFFOLD | $0.192 \pm 0.0018$ | $0.194 \pm 0.0025$ | $0.472 \pm 0.0016$ | $0.479 \pm 0.0034$ |
| FedBE | $0.182 \pm 0.0029$ | $0.189 \pm 0.0032$ | $0.440 \pm 0.0015$ | $0.463 \pm 0.0021$ |
| FedPA | $0.173 \pm 0.0031$ | $0.176 \pm 0.0020$ | $0.374 \pm 0.0033$ | $0.371 \pm 0.0025$ |
| FedEP | $0.121 \pm 0.0033$ | $\mathbf{0.118} \pm 0.0021$ | $0.289 \pm 0.0045$ | $0.273 \pm 0.0027$ |
| FALD | $0.135 \pm 0.0028$ | $0.127 \pm 0.0023$ | $0.267 \pm 0.0018$ | $0.269 \pm 0.0023$ |
| FedMDMI | $\mathbf{0.115} \pm 0.0019$ | $0.120 \pm 0.0031$ | $\mathbf{0.261} \pm 0.0023$ | $\mathbf{0.263} \pm 0.0029$ |

**Calibration and Uncertainty Estimation.** We evaluate the uncertainty estimates of all comparison methods, as shown in Table 3 and Figures 3(a) and 3(b). In most cases, our FedMDMI attains the lowest expected calibration error (ECE) and provides well-calibrated uncertainty estimates. Notably, the ECE of single point-estimation-based methods is consistently higher than that of posterior inference-based methods. While all these methods exhibit a tendency to be overconfident in their predictions, posterior inference demonstrates the potential to mitigate this overconfidence.

**Choice of appropriate $\alpha$ and $\beta$.** We have analyzed the impact of $\alpha$ that balances fitting and generalization in Section 5.2. We then analyze the hyperparameter $\beta$, which is used to estimate the mean of Gaussian prior and accelerate convergence. Due to space limit, we present these results in Appendix A.3.2, showing that even if $\beta = 0$, FedMDMI still achieves higher accuracy than FedAvg.

# 7 Conclusion

We have proposed a federated posterior inference approach, which mitigated bias in the posterior estimates and improved generalization by introducing a global model-data MI regularization. To approximate this global MI based on distributed data over the clients, we decomposed it into local MI terms. We showed that the optimal posterior of the local objective with MI regularization was a Gibbs posterior, which could be efficiently sampled by SGLD. We further provided a generalization analysis based on this global MI, and analyzed its impact on fitting and generalization in FL, enabling to present a federated version of the bias-variance trade-off.

# 8 Acknowledgement

This work was supported in part by the National Natural Science Foundation of China under Grant 62125109, Grant T2122024, Grant 62320106003, Grant 62371288, Grant 62431017, Grant 62401357, Grant 62401366, Grant 61931023, Grant 61932022, Grant 62120106007, and in part by the Program of Shanghai Science and Technology Innovation Project under Grant 24BC3200800.

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

# A  Appendix

## A.1  Detailed Algorithm Design

Here, we present a detailed learning procedure for our proposed FedMDMI, as outlined in Algorithm 1.

---

**Algorithm 1:** Proposed FedMDMI

---

**1** **Server Initialization**: $\Delta v_0 = 0$;
**2** **for** *each round $t = 1, 2, ...T$* **do**
**3** $\quad$ sample clients $\mathcal{P}_t \subseteq \mathcal{M}$
**4** $\quad$ **for** *each client $i \in \mathcal{P}_t$ in parallel* **do**
**5** $\quad\quad$ receive and initialize local model $w_{t,0}^i = w_t$
**6** $\quad\quad$ set $p(w) = \mathcal{N}\left(w \middle| w_t, \frac{1}{m}\sum_{k=1}^{K}\sqrt{2\eta_L^{t-1,k}\alpha}\mathbf{I}\right)$
**7** $\quad\quad$ **for** *each local step $k = 1, 2, \ldots, K$* **do**
**8** $\quad\quad\quad$ $U_i\left(w_{t,k-1}^i\right) = L_{S_i}(w_{t,k-1}^i) - \alpha \log p(w)|_{w=w_{t,k-1}^i}$
**9** $\quad\quad\quad$ $h_{t,k} \sim \mathcal{N}\left(\mathbf{0}, \sqrt{2\eta_L^{t,k}\alpha}\mathbf{I}\right)$
**10** $\quad\quad\quad$ $w_{t,k}^i = w_{t,k-1}^i - \eta_L^{t,k}\nabla U_i\left(w_{t,k-1}^i\right) + h_{t,k}$
**11** $\quad\quad$ **end**
**12** $\quad\quad$ $\Delta w_t^i = w_{t,K}^i - w_{t,0}^i$ and send $\Delta w_t^i$ to server
**13** $\quad$ **end**
**14** $\quad$ // at server:
**15** $\quad$ $\Delta w_t = \frac{1}{p}\sum_{i \in \mathcal{P}_t} \Delta w_t^i$
**16** $\quad$ $\Delta v_t = \frac{1}{1-\beta^t}\left(\beta \Delta v_{t-1} + (1-\beta)\Delta w_t\right)$
**17** $\quad$ $w_{t+1} = w_t + \eta \Delta v_t$
**18** $\quad$ broadcast $w_{t+1}$
**19** **end**

---

## A.2  Detailed Experiment Setup

The models, data splitting methods, and hyper-parameter settings employed in this paper closely adhere to those specified in the previous empirical benchmarks, which will be detaied in the following.

### A.2.1  Datasets

We utilize the visual datasets, including CIFAR-10 and CIFAR-100, and the language dataset Shakespeare. These datasets comprise 10, 100, and 80 different labels, respectively. The train and test splits for CIFAR-10, CIFAR-100, and Shakespeare are detailed in Table 4.

Table 4: Train and test splits.

| Dataset | No. Train | No. Test | No. Train per client (100 clients) | Batch size | Rounds |
|---|---|---|---|---|---|
| CIFAR-10 | 50000 | 10000 | 500 | 100 | 4000 |
| CIFAR-100 | 50000 | 10000 | 500 | 100 | 4000 |
| Shakespeare | 200000 | 40000 | 2000 | 100 | 1000 |

To generate heterogeneous splits for CIFAR-10 and CIFAR-100, we adopt a method where the training samples are divided by classes and assigned to clients. This approach aligns with the methodology used in FedDyn [Acar et al., 2021]. Specifically, we employ a Dirichlet distribution over the labels of the dataset to create a federated dataset. In this process, a vector of the same size as the number of classes is generated for each client using the Dirichlet distribution. These vectors represent the class

priority of each client. Subsequently, a label is sampled for each client based on these vectors, and images are sampled according to the selected label without replacement. This process is repeated until all the data are assigned to clients. It is important to highlight that the factor of Dirichlet distribution corresponds to the degree of data heterogeneity. For a Dirichlet factor of 0.2, each client has 80% of samples predominantly belonging to three or four different classes. And for a Dirichlet factor of 0.7, each client has 80% of samples mainly belonging to five or six different classes.

For Shakespeare, we employ LEAF to generate heterogeneous datasets by limiting the number of data points per client to 2000, a methodology akin to FedDyn [Acar et al., 2021]. In the heterogeneous splits, each client is associated with a speaking role comprising a few lines.

### A.2.2 Models

For CIFAR-10 and CIFAR-100, we employ a CNN model comprising two convolutional layers with sixty four $5 \times 5$ filters, two $2 \times 2$ max pooling layers, two fully connected layers with 384 and 192 neurons, and a softmax layer. A comprehensive description of the model is available in Table 5. Our CNN model closely resembles those utilized in FedAvg [McMahan et al., 2017] and FedDyn [Acar et al., 2021].

For Shakespeare, we utilize an RNN that takes a sequence of characters as input, embedding each into an 8-dimensional learned space. This sequence is then processed through a two-layer LSTM with a hidden size of 100 nodes, followed by a Softmax layer. A comprehensive description of the model is presented in Table 6. This neural network model is identical to the one utilized in FedProx [Li et al., 2020] and FedDyn [Acar et al., 2021].

Table 5: CNN Architecture for CIFAR-10 & CIFAR-100.

| Layer Type | Size |
|---|---|
| Convolution + ReLu | $5 \times 5 \times 64$ |
| Max Pooling | $2 \times 2$ |
| Convolution + ReLu | $5 \times 5 \times 64$ |
| Max Pooling | $2 \times 2$ |
| Fully Connected + ReLU | $1600 \times 384$ |
| Fully Connected + ReLU | $384 \times 192$ |
| Fully Connected | $192 \times 10$ & $192 \times 100$ |

Table 6: Shakespeare model architecture.

| Layer Type | Size |
|---|---|
| Embedding | (80, 8) |
| LSTM | (80, 100) |
| LSTM | (80, 100) |
| Fully Connected | (100, 80) |

### A.2.3 Hyper-Parameters

All different comparison approaches are implemented in PyTorch 1.4.0 and CUDA 9.2, with GEFORCE GTX 1080 Ti throughout our experiments. We tune the hyper-parameter over a grid search to compare the performance of different methods. For all these methods, we set the local learning rate as $0.1$, turn the local learning rate decay over $\{0.9999, 0.9995, 0.999, 0.995, 0.992\}$. Note that when aggregating model changes on the server, we introduce a global learning rate, and conduct a grid search for this global learning rate over the set $\{1, 3, 6, 9, 12, 15, 18\}$ for all methods. Moreover, we set up 5 epochs of local updates for CIFAR-10 and CIAR-100, 10 epochs of local updates for Shakespeare. We also set $\beta = 0.9$ for the momentum moving average used in the global aggregation step of FedM, FedPA, FedEP, FALD, and FedMD. For our FedMD, we turn $\alpha$ over $\{10^{-6}, 10^{-7}, 10^{-8}, 10^{-9}, 10^{-10}\}$. For FedEP, we turn $\alpha_{cov}$ over $\{10^{-2}, 10^{-3}, 10^{-4}, 10^{-5}, 10^{-6}\}$. Note that in FALD, the original work employs an extremely small learning rate ($10^{-9}$), leading to exceedingly slow convergence. In this paper, we utilize a larger learning rate and introduce an noise-scale factor to control the noise. For FedBE, We focus on Gaussian and allocate 10% of the training data of each client to be available at the server.

### A.3 Additional Experimental Results

### A.3.1 Calibration and Uncertainty Estimation for Other Settings

Here, we present the additional experimental results on uncertainty estimates, as detailed in Table 7.

Table 7: ECE (averaged over 5 random seeds).

| Method | CIFAR10 | | CIFAR100 | |
|---|---|---|---|---|
| | Dir (0.2)-H | Dir (0.7)-H | Dir (0.2)-H | Dir (0.7)-H |
| FedAvg | 0.168 | 0.170 | 0.434 | 0.440 |
| FedM | 0.164 | 0.162 | 0.471 | 0.461 |
| MimeLite | 0.177 | 0.186 | 0.469 | 0.478 |
| SCAFFOLD | 0.186 | 0.195 | 0.480 | 0.483 |
| FedPA | 0.167 | 0.174 | 0.382 | 0.380 |
| FedEP | 0.130 | 0.124 | 0.298 | 0.284 |
| FALD | 0.139 | 0.126 | 0.274 | **0.259** |
| FedMDMI | **0.125** | **0.122** | **0.267** | 0.262 |

### A.3.2 Experimental Results Under Different $\beta$

In this section, we provide the test accuracy of the proposed FedMDMI vs. FedAvg, under different choices of $\beta$, as shown in Figures 4(a)-4(c).

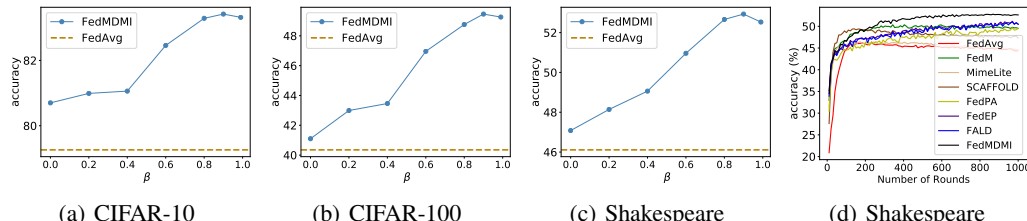

| (a) CIFAR-10 | (b) CIFAR-100 | (c) Shakespeare | (d) Shakespeare |

Figure 4: (a), (b) and (c): sensitivity analysis of $\beta$; and (d): test accuracy of clients vs. number of communication rounds achieved by different comparison methods.

### A.3.3 Convergence Curve for Shakespeare

In this section, we provide the convergence curves for different methods on the Shakespeare dataset, as shown in Figure 4(d).

### A.3.4 Other Experimental Results with More Settings and Model Structures

**Extension to a PFL Context:** The survey paper [Cao et al., 2023] provides a overview of Bayesian federated learning (BFL), including the client- and server-side BFL and various other categories of BFL. For example, FedHB [Kim and Hospedales, 2023] proposes a hierarchical BFL approach, using hierarchical Bayesian modeling to describe the generative process of clients' local data with local models governed by a higher-level global variate. FedSI [Chen et al., 2024] proposes a personalized BFL approach, performing posterior inference (i.e., Linearized Laplace Approximation) over an individual subset of network parameters for each client, while keeping other parameters deterministic. Inspired by FedRep [Collins et al., 2021] and FedSI [Chen et al., 2024], we consider applying our algorithm to personalized Bayesian FL, named PerFed-MDMI. FedRep learns a shared data representation across clients and unique local heads for each client to fulfill their personal objectives. FedSI further updates the distributions of model parameters over the representation layers and sends these updated distributions to the server for global aggregation throughout the entire training. Model parameters of the decision layers are then fine-tuned during the evaluation phase. For subnetwork (i.e., representation layers) inference, instead of using the Linearized Laplace Approximation from FedSI, PerFed-MDMI employs our model-data mutual information regularization technique. Additionally, since our posterior inference method does not involve covariance matrix estimation, we do not consider selecting a smaller subnetwork to reduce computational and storage overhead.

For experimental comparison, we first evaluate our algorithm against FedProx and FedHB on a traditional FL task, namely global prediction. Results in Table 8 demonstrate that our algorithm generally outperforms FedProx and FedHB.

For the personalized FL task, we compare our newly designed algorithm, PerFed-MDMI, with FedAvg and FedRep, demonstrating a clear improvement in Table 9, where all data at each client is split into 70% training set and 30% test set. This indicates that our mutual information regularization

Table 8: Global Performance comparison under various settings.

| Dataset | Setting | FedPorx | FedHB-NIW | FedMDMI (Ours) |
|---|---|---|---|---|
| CIFAR-10 | Dir (0.2)-L | 79.82 | 83.19 | **83.42** |
|  | Dir (0.7)-L | 80.59 | 83.25 | **83.81** |
|  | Dir (0.2)-H | 79.93 | 83.28 | **83.56** |
|  | Dir (0.7)-H | 80.88 | **83.80** | 83.76 |
| CIFAR-100 | Dir (0.2)-L | 42.89 | 48.15 | **49.46** |
|  | Dir (0.2)-H | 44.12 | 49.26 | **50.45** |
|  | Dir (0.2)-H | 43.07 | 49.02 | **49.70** |
|  | Dir (0.7)-H | 44.85 | 49.38 | **50.71** |

Table 9: Personalized Performance comparison under various settings.

| Dataset | Setting | FedAvg | FedRep | FedPer-MDMI (Ours) |
|---|---|---|---|---|
| CIFAR-10 | Dir (0.2)-L | 75.46 | 76.78 | **79.87** |
|  | Dir (0.7)-L | 76.89 | 77.45 | **80.49** |
|  | Dir (0.2)-H | 75.98 | 77.08 | **80.45** |
|  | Dir (0.7)-H | 77.10 | 78.97 | **80.90** |
| CIFAR-100 | Dir (0.2)-L | 31.02 | 40.52 | **48.08** |
|  | Dir (0.2)-H | 32.87 | 41.87 | **48.76** |
|  | Dir (0.2)-H | 33.24 | 42.05 | **48.20** |
|  | Dir (0.7)-H | 33.88 | 42.74 | **49.42** |

technique is compatible with FedRep and FedSI, effectively promoting personalized FL. Additionally, since our focus in this paper is on training a global posterior rather than the personalized posteriors, we did not directly compare the accuracy with FedSI, FedPop, and other baselines in personalized FL due to the time constraint.

**Influence of the Degree of Data Heterogeneity:** We also conduct additional experiments, observing that as the degree of heterogeneity increases, the performance of all the comparison algorithms declines, as shown in 10. However, the performance of our FedMDMI decreases to a lesser extent in the most cases. This indicates that our FedMDMI is more robust to the increasing heterogeneity as compared to the other baselines. Since we cannot display images here, we present these results in the following table form for now, and we will convert it to line plots in the final version of our manuscript.

We have not yet reached a conclusion on how data heterogeneity affects the ECE, as shown in 11. It is clear that uncertainty estimation is influenced by local sample size, and smaller sample sizes tend to lead to overconfident decisions. On the other hand, data heterogeneity has a more pronounced effect on accuracy.

Table 10: Test Accuracy (%) v.s. Data heterogeneity.

| Method | CIFAR10 | | | | CIFAR100 | | | |
|---|---|---|---|---|---|---|---|---|
|  | iid-H | Dir (0.7)-H | Dir (0.2)-H | Dir (0.1)-H | iid-H | Dir (0.7)-H | Dir (0.2)-H | Dir (0.1)-H |
| FedAvg | 83.22 | 80.31 | 79.68 | 77.96 | 48.65 | 42.35 | 40.32 | 38.14 |
| FedBE | 84.06 | 82.33 | 81.25 | 79.19 | 51.68 | 46.29 | 44.82 | 42.06 |
| FedPA | 84.82 | 82.93 | 82.78 | 80.37 | 52.44 | 49.66 | 48.51 | 46.79 |
| FedEP | 84.93 | **83.79** | 83.30 | 81.23 | 52.99 | 50.02 | 49.08 | 47.32 |
| FedMDMI | **85.05** | 83.76 | **83.56** | **82.28** | **53.28** | **50.71** | **49.70** | **48.41** |

Table 11: ECE v.s. Data heterogeneity.

| Method | CIFAR10 | | | | CIFAR100 | | | |
|---|---|---|---|---|---|---|---|---|
|  | iid-H | Dir (0.7)-H | Dir (0.2)-H | Dir (0.1)-H | iid-H | Dir (0.7)-H | Dir (0.2)-H | Dir (0.1)-H |
| FedAvg | 0.174 | 0.170 | 0.168 | 0.160 | 0.437 | 0.440 | 0.434 | 0.430 |
| FedBE | 0.181 | 0.184 | 0.187 | 0.169 | 0.452 | 0.467 | 0.438 | 0.442 |
| FedPA | 0.179 | 0.174 | 0.167 | 0.168 | 0.387 | 0.380 | 0.382 | 0.361 |
| FedEP | 0.137 | 0.124 | 0.130 | 0.125 | 0.281 | 0.284 | 0.298 | 0.277 |
| FedMDMI | **0.133** | **0.122** | **0.125** | **0.117** | **0.272** | **0.262** | **0.267** | **0.259** |

Note Within the comparison methods, FedEP utilize the variational approximation to obtain the global posterior. However, determining whether MCMC or VA is more suitable for federated posterior inference is in general a complex problem, as their effectiveness depends on the specific context. Due to the data heterogeneity issue, regularization at the client level is often necessary to mitigate bias in the local posteriors. Our FedMDMI employs the model-data mutual information regularization alongside the MCMC for posterior inference. In contrast, FedEP uses the posterior from the previous round to regularize the current local posterior. However, in some practical federated scenarios with low client participation rates, this retained local posterior in FedEP can become very stale, thus reducing its effectiveness in addressing the data heterogeneity issue. This also contributes to the less stable and slower convergence observed for FedEP compared to our FedMDMI.

**Influence of Model Architectures:** We evaluate the performance of our FedMDMI by using ResNet18 on the CIFAR-10 and CIFAR-100 datasets. Table 12 shows that our FedMDMI still outperforms other comparison methods in terms of the generalization performance, with the ResNet18 architecture. Here, we replace the batch norm with group norm, and set the number of clients to 20, such that the 10% and 5% sampling rates correspond to only two and one client participating in training per communication round, respectively. This may highlight the robustness of our FedMDMI to some possible model-architecture changes and its ability to adapt to various models.

Table 12: Performance comparison on Resent-18 under various settings.

| Dataset | Setting | FedAvg | FedM | MimeLite | SCAFFOLD | FedBE | FedPA | FedEP (I) | FALD | FedMDMI (Ours) |
|---|---|---|---|---|---|---|---|---|---|---|
| | | | | | Top-1 Test Accuracy (%). | | | | | |
| CIFAR10 on Resent18 | Dir (0.2)-L | 85.12 | 86.91 | 85.40 | 87.01 | 86.98 | 86.49 | 87.05 | 87.14 | **87.86** |
| | Dir (0.7)-L | 85.41 | 87.29 | 85.71 | 87.10 | 87.35 | 87.91 | 87.46 | 87.01 | **87.82** |
| | Dir (0.2)-H | 85.69 | 87.32 | 85.16 | 87.24 | 87.11 | 87.21 | 87.18 | 87.09 | **87.92** |
| | Dir (0.7)-H | 86.20 | 87.47 | 86.09 | 87.98 | 87.51 | 87.56 | 88.06 | 87.92 | **88.25** |
| CIFAR100 on Resent18 | Dir (0.2)-L | 53.19 | 55.34 | 53.28 | 57.21 | 57.04 | 57.08 | 57.36 | 57.28 | **58.19** |
| | Dir (0.7)-L | 53.40 | 55.80 | 53.76 | 57.64 | 57.46 | 57.16 | 57.92 | 57.16 | **58.26** |
| | Dir (0.2)-H | 53.40 | 55.49 | 53.46 | 57.38 | 57.16 | 57.22 | 57.40 | 57.32 | **58.24** |
| | Dir (0.7)-H | 53.91 | 56.03 | 54.16 | 57.98 | 57.81 | 58.24 | 58.13 | 57.59 | **58.81** |

**Influence of the Number of Local Data Samples:** We conduct additional experiments to analyze the impact of the number of local data samples. Given that the total number of data samples in CIFAR-10 and CIFAR-100 is fixed, we controlled the number of samples on each client by adjusting the number of clients. With fewer data samples on a client, the local model is more prone to overfitting. The results are shown in Table 13. Our FedMDMI maintains superior performance even with scarce local data, demonstrating that our MI regularization-based posterior estimation effectively alleviates the overfitting caused by data scarcity.

Table 13: Performance Comparison across various number of data sample on clients

| Dataset | Setting | FedAvg | FedM | MimeLite | SCAFFOLD | FedBE | FedPA | FedEP (I) | FALD | FedMDMI (Ours) |
|---|---|---|---|---|---|---|---|---|---|---|
| | | | | | Top-1 Test Accuracy (%). | | | | | |
| CIFAR10 (Dir (0.2)-L) | 50 | 77.02 | 80.08 | 77.15 | 79.25 | 78.99 | 80.42 | 80.34 | 80.59 | **80.86** |
| | 100 | 77.51 | 80.49 | 77.60 | 79.57 | 79.05 | 80.97 | 81.09 | 80.97 | **81.47** |
| | 250 | 78.49 | 81.78 | 78.48 | 80.97 | 80.25 | 81.33 | 82.15 | 81.72 | **82.87** |
| | 500 | 79.26 | 82.12 | 79.54 | 82.11 | 81.49 | 82.29 | 82.63 | 82.56 | **83.42** |
| CIFAR100 (Dir (0.2)-L) | 50 | 36.76 | 40.49 | 35.87 | 42.24 | 38.84 | 42.26 | 43.15 | 42.14 | **43.84** |
| | 100 | 37.05 | 42.86 | 37.88 | 44.41 | 39.71 | 44.81 | 45.41 | 43.89 | **46.08** |
| | 250 | 39.61 | 45.62 | 39.15 | 46.97 | 43.08 | 46.80 | 47.15 | 45.79 | **47.90** |
| | 500 | 40.35 | 47.13 | 40.25 | 47.76 | 44.55 | 48.59 | 48.89 | 48.43 | **49.46** |

**Influence of Batch Size:** For the batch size, we also conduct additional experiments to demonstrate the effect of batch size on the performance. The results are shown in Table 14. Our method demonstrates a greater robustness across different batch sizes compared to other baselines.

Table 14: Performance Comparison across various batch-size for local optimaization

| Dataset | Setting | FedAvg | FedM | MimeLite | SCAFFOLD | FedBE | FedPA | FedEP (I) | FALD | FedMDMI (Ours) |
|---|---|---|---|---|---|---|---|---|---|---|
| | | | | | Top-1 Test Accuracy (%). | | | | | |
| CIFAR10 (Dir (0.2)-L) | 5 | 73.54 | 76.78 | 73.84 | 77.89 | **79.23** | 78.43 | 78.45 | 77.05 | 77.45 |
| | 50 | 79.26 | 82.12 | 79.54 | 82.11 | 81.49 | 82.29 | 82.63 | 82.56 | **83.42** |
| | 100 | 79.18 | 82.08 | 79.80 | 82.02 | 81.30 | 82.18 | 82.41 | 82.29 | **83.49** |
| | 200 | 79.05 | 81.49 | 78.78 | 81.83 | 80.16 | 81.28 | 80.97 | 80.78 | **81.91** |
| CIFAR100 (Dir (0.2)-L) | 5 | 33.97 | 42.54 | 34.90 | 42.55 | **43.97** | 41.43 | 42.63 | 41.01 | 41.45 |
| | 50 | 40.35 | 47.13 | 40.25 | 47.76 | 44.55 | 48.59 | 48.89 | 48.43 | **49.46** |
| | 100 | 40.28 | 46.91 | 40.46 | 48.12 | 44.81 | 47.34 | 48.45 | 48.64 | **49.04** |
| | 200 | 39.06 | 43.67 | 38.34 | 45.49 | 44.73 | 45.56 | 45.54 | 45.34 | **46.43** |

**Extension to Other Dataset:** We further used TensorFlow Federated to generate the Stack Overflow dataset. Similar to Cheng et al. [2023], due to the limited graphics memory and rebuttal-time, we utilize only a sample of 300 clients from the original dataset, with the following observations.

Table 15: Performance comparison on Stack Overflow under various settings.

| | | | | | | | | | |
|---|---|---|---|---|---|---|---|---|---|
| | | Top-1 Test Accuracy (%). | | | | | | | |
| Dataset | Setting | FedAvg | FedM | MimeLite | SCAFFOLD | FedPA | FedEP (I) | FALD | FedMDMI (Ours) |
| Stack Overflow | non-iid-L | 40.08 | 41.24 | 40.16 | 42.11 | 42.66 | 42.79 | 42.34 | **43.00** |
| | non-iid-H | 41.23 | 42.01 | 41.34 | 42.50 | 42.67 | 43.01 | 42.41 | **43.11** |

As shown in Table 15, on the Stack Overflow dataset, our FedMDMI continues to outperform the majority of other optimization algorithms designed to address the data heterogeneity. This can be attributed to our proposed model-data mutual information regularization that enhances generalization.

**Experimental Results of Complexity Analysis:** To provide empirical evidence on the complexity analysis, we conduct experiments to evaluate the running time of each communication round for these algorithms. Taking CIFAR100 dataset with participation rate $\frac{p}{m} = 0.1$ as an example, the average time required to execute a communication round for different models LeNet and ResNet-18 with GEFORCE GTX 1080 Ti is as follows:

For LeNet: -FedAvg: 5.29 seconds; - FedPA: 6.94 seconds; - FedEP: 11.25 seconds; - FedBE: 17.36 seconds; - FedMDMI: 5.36 seconds.

For ResNet-18: - FedAvg: 13.05 seconds; - FedPA: 16.87 seconds; - FedEP: 25.10 seconds; - FedBE: 43.06 seconds; - FedMDMI: 13.88 seconds.

This shows that the time consumed by our algorithm per communication round does not exhibit a significant difference compared to FedAvg. The time required by FedBE is much larger than that of other algorithms. Moreover, as the network size increases, the time taken by all baselines also increases, which aligns with our intuition.

### A.3.5 Other Experimental Results with $\beta$-PredBayes

Table 16: ECE (averaged over 5 random seeds).

| Method | CIFAR-10 | | | | CIFAR-100 | | | |
|---|---|---|---|---|---|---|---|---|
| | Dir (0.2)-L | Dir (0.7)-L | Dir (0.2)-H | Dir (0.7)-H | Dir (0.2)-L | Dir (0.7)-L | Dir (0.2)-H | Dir (0.7)-H |
| FedAvg | 0.169 | 0.165 | 0.168 | 0.170 | 0.429 | 0.432 | 0.434 | 0.440 |
| $\beta$-PredBayes | 0.124 | 0.126 | 0.124 | 0.126 | 0.262 | **0.242** | **0.260** | 0.266 |
| FedMDMI (Ours) | **0.115** | **0.120** | **0.125** | **0.122** | **0.261** | 0.263 | 0.267 | **0.262** |

We present the experimental results of our algorithm compared with the additional baseline, $\beta$-PredBayes [Hasan et al., 2024]. $\beta$-PredBayes is designed to provide well-calibrated uncertainty estimates other than a good prediction accuracy. In addition, $\beta$-PredBayes is also designed for a single round of communication, making it unfair to directly compare its accuracy with other algorithms. Moreover, the $\beta$-PredBayes requires auxiliary data for the knowledge distillation at the server. To meet this requirement, we allocate 10% of the training data of each client to be available at the server.

Experimental results in Table 16 show that our algorithm outperforms $\beta$-PredBayes in terms of the ECE in most cases. It is worth noting that enhancing the quality of the auxiliary data could improve performance of these algorithms.

We note that $\beta$-PredBayes sometimes performs better for CIFAR-100, we consider that there are two possible reasons. *i*) In a federated heterogeneous scenario involving CIFAR-100, which comprises 100 categories compared to CIFAR-10, each client typically handles a subset of 13-16 (or 20-25) categories when setting $Dir = 0.2$ (or $Dir = 0.7$). Consequently, with such a high degree of heterogeneity, clients' training tends to incline towards overconfident predictions, possibly neglecting some categories entirely. Then, when performing the aggregation globally (e.g., in our FedMDMI algorithm), it may produce overconfident predictions. *ii*) $\beta$-PredBayes additionally utilizes a subset of the training data encompassing all categories to perform the global knowledge distillation (which may though incur the additional storage/computation and privacy leakage issues at server). This strategy significantly mitigates the overconfident predictions stemming from the training on locally heterogeneous data.

### A.4  Proof

#### A.4.1  Proof of Global Posterior Decomposition

**Global posterior decomposition** is a widely adopted approach in federated Bayesian inference Al-Shedivat et al. [2020] and embarrassingly parallel (EP)-MCMC Neiswanger et al. [2014], which decomposes the global posterior into a product of client posteriors:

$$
p(w|S) \propto p(S|w)p(w) \overset{(A_1)}{=} \left\{ \prod_{i=1}^{m} p\left(S_i|w\right) \right\} p(w) \propto \left\{ \prod_{i=1}^{m} \frac{p\left(w|S_i\right)}{p_i(w)} \right\} p(w)
$$
$$
= \left\{ \prod_{i=1}^{m} p\left(w|S_i\right) \right\} \left\{ \frac{p(w)}{\prod_{i=1}^{m} p_i(w)} \right\}.
$$
(21)

where (A1) is due to the assumption of the global likelihood being conditionally independent given $w$ Al-Shedivat et al. [2020], Guo et al. [2023], Neiswanger et al. [2014], and $\tau \triangleq \frac{p(w)}{\prod_{i=1}^{m} p_i(w)}$ represents the ratio of the global prior to the product of client priors. Note that there exists another posterior decomposition Sefidgaran et al. [2023]which assumes independence among the local posteriors given the global model of previous communication round, rather than based on the Bayesian inference principle, as shown in the following section.

#### A.4.2  Difference between the Bayesian Inference Approach and Model Distribution Update over Communication Rounds for FL

There is an another posterior decomposition, based on the Model Distribution Update over communication rounds [Chor et al., 2023, Sefidgaran et al., 2023]. To mitigate confusion, we will delineate the difference between the "Model Distribution Update over communication rounds" and "Bayesian Inference" for FL.

***i*) Posterior Decomposition based on Model Distribution Update over Communication Rounds [Chor et al., 2023, Sefidgaran et al., 2023]**

In this context, along this update procedure, the training process of FL is viewed as a **Markov process** indexed by time $t$. That is, at each communication round $t$, $w_t^i$ is regarded as a random variable, determined by the previous round's random variable $w_{t-1}$ (initialization of the global model) and the current round's state $S_i^t$, i.e., $p\left(w_t^i|S_i^t, w_{t-1}\right)$. By defining $\mathbf{W} = \left(w_{[T]}^{[M]}, w_{[T]}\right)$, which denotes the set of random variables encompasses model from all clients [M] and server across all communication rounds [T] , we directly have

$$
p(\mathbf{W}|S) = \prod_{t \in [T]} \left\{ \prod_{i \in [M]} p\left(w_t^i|S_i^t, w_{t-1}\right) p(w_t|w_t^{[M]}]) \right\}.
$$
(22)

This equation corresponds to Eq. (2) in Sefidgaran et al. [2023], where $w_t$ and $w_t^i$ are the random variables rather than samples. It is crucial to note its primary distinction from Eq. (21) in our posterior decomposition: this equation directly assumes independence among the posteriors $p(w_t^i|S_i)$ given $w_{t-1}$ (the initial model at current round), without relying on Bayes' theorem or the assumption of likelihood independence given $w$. Additionally, it is worth mentioning that the distribution $p(w_t)$ of the global model varies over time.

***ii*) Global posterior decomposition based on Bayesian inference**

On the contrary, our entire work is based on Bayesian inference. In our context, the parameters of a learned global model $w$ are treated as random variables rather than single points, then we aim to estimate the global posterior $p(w|S)$ of model weights $w$ given the global dataset $S$. Due to the inability to exchange raw client data in FL, we decompose the global posterior into a product of clients' posteriors:

$$
p(w|S) \propto \prod_{i=1}^{m} p\left(w|S_i\right),
$$
(23)

which is based on the assumption of the global likelihood being conditionally independent given $w$, i.e., $p(S_1, \ldots, S_m|w) = \prod_{i=1}^{m} p(S_i|w)$.

This equation indicates that under the independence assumption of likelihood and prior assumption, the posterior distribution of the full data $S$ is the product of local posterior $p(w|S_i)$. Consequently, our subsequent objective is to estimate the lcoal posterior $p(w|S_i)$ and then approximate global $p(w|S)$. Alternative methods include obtaining samples drawn from the posterior (MCMC), or to fit the optimal distribution by Gaussian assumption (Variational Inference).

Specifically, in our work, following the introduction of our newly proposed Model-Data MI regularization, there is an **optimal global posterior** $p(w|S_i)$, also referred to as the Gibbs posterior. To obtain samples form global posterior $p(w|S) = \prod_{i=1}^{m} p(w|S_i)$ through optimization (or training iteration with $t$), we employ the MCMC sampling (SGLD). After $t$ communication rounds and $k$ local iterations, $w_{t,k}^i$ denotes a locally sampled model of client $i$, following the distribution $p(w|S_i)$, and $w_t$ denotes a globally aggregated sample, following the distribution $p(w|S)$.

### *iii*) Further clarification

In summary, we would like to highlight the difference between "Bayesian Inference" and "Model Distribution Update over Communication Rounds" for FL from the following three perspectives.

- First, within the context of our work, there is an estimated posterior $p(w|S)$ (i.e., Gibbs posterior) when given a task $S$ and prior assumption. We use the sample $w_t$ to approximate sample from this posterior through the MCMC sampling methods. Conversely, within the context of "Model Distribution Update over Communication Rounds", the distribution of a global model $p(w_t|S)$ evolves with time $t$.

- Second, within the context of our work, $w$ is a **random variable** of learned global model, while $w_t$ and $w_{t,k}^i$ represent **samples** obtained through the MCMC sampling, following the distribution $p(w|S)$ and $p(w|S_i)$, respectively. Consequently, the notation $p(w_{t,k}^i|S_i, w_{t-1})$ is **inaccurate and meaningless** in our context. Conversely, within the context of "Model Distribution Update over Communication Rounds", $w_{t,k}^i$ and $w_t$ are both **random variables** at time $t$, and $p(w_{t,k}^i|S_i, w_{t-1})$ holds **true**.

- Third, the global posterior decomposition in our work does not directly assume independence among the posteriors $p(w|S_i)$. Instead, it relies on the conditional independence of the likelihood function and prior assumptions. Conversely, within the context of "Model Distribution Update over Communication Rounds", the posterior decomposition directly assumes independence among the posteriors $p(w_t^i|S_i)$ given $w_{t-1}$.

Thus, literature [Chor et al., 2023, Sefidgaran et al., 2023] offers a new and intriguing direction for analyzing the relationship between the generalization bound and number of communication rounds $t$. However, it is worth noting that these analyses fundamentally diverge from the focus of our work, which is in essence a Bayesian inference-based FL algorithm.

### A.4.3   Proof of Global Model-Data MI Decomposition

**Proposition A.1.** *(Global Model-Data MI Decomposition). Suppose that $S = \{S_1, \ldots, S_m\}$ consists of data from $m$ clients, then based on the chain rule of mutual information, we have:*

$$
\begin{aligned}
I(w; S) &= \sum_{i=1}^{m} \left[ I\left(w; S_i\right) - I\left(S_i; S^{i-1}\right) \right] \\
&\leq \sum_{i=1}^{m} I\left(w; S_i\right),
\end{aligned}
\tag{24}
$$

*where $S^{i-1} \triangleq \{S_1, \ldots, S_{i-1}\}$.*

*Proof.* To improve readability, we first present the typical chain rule for mutual information here. For any random variables $x$, $y$ and $z$, we have $I(x, y; z) = I(y; z) + I(x; z|y)$. By iteratively applying this chain rule of mutual information, we have:

$$I(w; S) = \sum_{i=1}^{m} I\left(w; S_i | S^{i-1}\right),$$

where $S^{i-1} \triangleq \{S_1, \ldots, S_{i-1}\}$. Then, based on the assumption of the global likelihood being conditionally independent given $w$, we have

$$
\begin{aligned}
I\left(w; S_i | S^{i-1}\right) &= I\left(w; S_i | S^{i-1}\right) + I\left(S_i; S^{i-1}\right) - I\left(S_i; S^{i-1}\right) \\
&\overset{(A_1)}{=} I\left(w, S^{i-1}; S_i\right) - I\left(S_i; S^{i-1}\right) \\
&\overset{(A_2)}{=} I\left(w; S_i\right) + I\left(S^{i-1}; S_i | w\right) - I\left(S_i; S^{i-1}\right) \\
&\overset{(A_3)}{=} I\left(w; S_i\right) - I\left(S_i; S^{i-1}\right) \\
&\leq I\left(w; S_i\right),
\end{aligned}
\tag{25}
$$

where (A1) and (A2) are due to the the chain rule of mutual information, and (A3) is from the assumption of the global likelihood being conditionally independent given $w$ Al-Shedivat et al. [2020], Guo et al. [2023]. That is,

$$I\left(S^{i-1}; S_i | w\right) = \mathbb{E}_{p(w)}\left[\mathrm{KL}\left(p(S^{i-1}, S_i | w) \| p(S^{i-1} | w) p(S_i | w)\right)\right] = 0. \tag{26}$$

$\square$

### A.4.4 Proof of Lemma 4.2

**Lemma A.2.** *Xu and Raginsky [2017] The Gibbs posterior is the minimizer of the objective for client update in Eq. (9):*

$$p\left(w | S_i\right) = \frac{1}{B_i} \exp\left[-\frac{1}{\alpha}\left(L_{S_i}(w) - \alpha \log p_i(w)\right)\right], \tag{27}$$

*where $B_i$ is a normalization factior.*

*Proof.* In order for our paper to be self-contained, we re-state the proof in Xu and Raginsky [2017], Wang et al. [2021b] here.

For the local objective, we have:

$$
\begin{aligned}
\min_{p(w|S_i)} F_i(w) &= \mathbb{E}_{p(w|S_i)}\left[L_{S_i}(w)\right] + \alpha I(w; S_i) \\
&= \int p(w | S_i)\left[L_{S_i}(w)\right] dw + \alpha \int \int p(w, S_i)[\log p(w | S_i) - \log p_i(w)] dw dS_i.
\end{aligned}
\tag{28}
$$

Consequently, differentiating $F_i(w)$ w.r.t. $p(w|S_i)$ results in:

$$\nabla_{p(w|S_i)} F_i(w) = L_{S_i}(w) + \alpha \log p\left(w | S_i\right) - \alpha \log p_i(w) + \alpha.$$

Setting $\nabla_{p(w|S_i)} F_i(w) = 0$, we then have:

$$
\begin{aligned}
\log p\left(w | S_i\right) &= -\frac{1}{\alpha} L_{S_i}(w) + \log p_i(w) - 1 \\
p\left(w | S_i\right) &= p_i(w) \exp\left\{-\frac{1}{\alpha} L_{S_i}(w)\right\} \exp\{-1\} \\
p\left(w | S_i\right) &\propto p_i(w) \exp\left\{-\frac{1}{\alpha} L_{S_i}(w)\right\} \\
&\propto \exp\left\{-\frac{1}{\alpha}\left[L_{S_i}(w) - \alpha \log p_i(w)\right]\right\}.
\end{aligned}
\tag{29}
$$

To integrate the distribution to 1, we add an additional normalization factor $B_i$. $\square$

### A.4.5 Generalization Analysis

**Theorem A.3.** *(Generalization Bounds for FL). Suppose that $\ell_i(w, z_i^j)$ for all $i \in \mathcal{M}$ is bounded by $C$ and independent, then the expected generalization error statisfies:*

$$\mathbb{E}\left[L_\mathcal{P}(w) - L_\mathcal{S}(w)\right] \leq \sqrt{\frac{C^2 I(w; S)}{2mn}} + \sqrt{\frac{C^2 I(w; D)}{2m}}, \tag{30}$$

*where $m$ is the number of clients, $n$ is the number of samples on the client (assuming, without loss of generality, that the number of samples for all clients is equal), and $D$ is the set of distributions $p(S_i)$.*

In the subsequent proof, we primarily leverage the theorem and lemma drawn from PAC-Bayesian learning in Alquier [2021]. Note that while the derivation about the model-data mutual information for centrilized learning in Xu and Raginsky [2017] is concise, it is only applicable to i.i.d. cases and may not be extended directly to federated heterogeneous scenarios.

*Proof.* Under the FL settings, it is crucial to consider not only the gaps arising from the unseen client data (i.e., participating error), but also the gaps stemming from the unseen client distributions (i.e., participation gap). Following the two-level framework proposed by Yuan et al. [2021] and Hu et al. [2023], we define the loss functions separately as:

1) the population risk:

$$L_\mathcal{P}(w) = \mathbb{E}_{p(S_i)\sim P} L_{p(S_i)}(w) = \mathbb{E}_{p(S_i)\sim P}\left[\mathbb{E}_{z_i\sim p(S_i)}[\ell_i(w, z_i)]\right]; \tag{31}$$

2) the semi-empirical risk:

$$L_\mathcal{D}(w) = \frac{1}{m}\sum_{i\in\mathcal{M}} L_{p(S_i)}(w) = \frac{1}{m}\sum_{i\in\mathcal{M}} \mathbb{E}_{z_i\sim p(S_i)}[\ell_i(w, z_i)]; \tag{32}$$

3) the empirical risk:

$$L_\mathcal{S}(w) = \frac{1}{m}\sum_{i\in\mathcal{M}} L_{S_i}(w) = \frac{1}{m}\sum_{i\in\mathcal{M}}\left[\frac{1}{n_i}\sum_{j=1}^{n_i}\ell_i(w, z_i^j)\right]. \tag{33}$$

Then, the generalization error can be written as:

$$\begin{aligned}
&\mathbb{E}\left[L_\mathcal{P}(w) - L_\mathcal{S}(w)\right]\\
=&\mathbb{E}_P\mathbb{E}_{p(S_i)}\mathbb{E}_{p(w|S)}\left[L_\mathcal{P}(w) - L_\mathcal{D}(w)\right] + \mathbb{E}_P\mathbb{E}_{p(S_i)}\mathbb{E}_{p(w|S)}\left[L_\mathcal{D}(w) - L_\mathcal{S}(w)\right]\\
=&\mathbb{E}_P\mathbb{E}_{p(S_i)}\mathbb{E}_{p(w|S)}\left[L_\mathcal{P}(w) - \frac{1}{m}\sum_{i\in\mathcal{M}} L_{p(S_i)}(w)\right] + \mathbb{E}_P\mathbb{E}_{p(S_i)}\mathbb{E}_{p(w|S)}\left[\frac{1}{m}\sum_{i\in\mathcal{M}}\left(L_{p(S_i)}(w) - L_{S_i}(w)\right)\right]
\end{aligned} \tag{34}$$

We first derive the upper bound of **the second term** in Eq. (34).

**Lemma A.4.** *(Generalization Bound for the second term). Let $p(S_i)$ be a distribution over examples in client $i$ and let $P$ be a mata distribution. Let $m$ be the number of clients and $n$ be the number of samples on the client. Then, we have:*

$$\mathbb{E}_P\mathbb{E}_{p(S_i)}\mathbb{E}_{p(w|S)}\left[\frac{1}{m}\sum_{i\in\mathcal{M}}\left(L_{p(S_i)}(w) - L_{S_i}(w)\right)\right] \leq \sqrt{\frac{C^2 I(w; S)}{2mn}}. \tag{35}$$

*Proof.* Note that Hoeffding's inequality only requires that the random variables be independent and not necessarily identically distributed. We apply Hoeffding's inequality to a single variable $U_1$ with values in the interval $[a, b]$ for the case of $n = 1$, which simply states that:

$$\mathbb{E}\left(e^{t[U_1 - \mathbb{E}(U_1)]}\right) \leq e^{\frac{t^2(b-a)^2}{8}}. \tag{36}$$

When $U_1 = \mathbb{E}_{z_i\sim p(S_i)}[\ell_i(w, z_i)] - \ell_i(w, z_i)$, we have:

$$\mathbb{E}_{p(S_i)}\left(e^{t\left[\mathbb{E}_{z_i\sim p(S_i)}(\ell_i(w, z_i)) - \ell_i(w, z_i)\right]}\right) \leq e^{\frac{t^2 C^2}{8}}. \tag{37}$$

That is:

$$
\mathbb{E}_P \mathbb{E}_{p(S_i)} \left( e^{t \sum_{i=1}^{m} \sum_{j=1}^{n} \left[ \mathbb{E}_{z_i^j \sim p(S_i)} \left( \ell_i(w, z_i^j) \right) - \ell_i(w, z_i^j) \right]} \right)
$$

$$
= \mathbb{E}_P \mathbb{E}_{p(S_i)} \left( \prod_{i=1}^{m} \prod_{j=1}^{n} e^{t \left[ \mathbb{E}_{z_i^j \sim p(S_i)} \left( \ell_i(w, z_i^j) \right) - \ell_i(w, z_i^j) \right]} \right)
$$

$$
= \prod_{i=1}^{m} \prod_{j=1}^{n} \mathbb{E}_P \mathbb{E}_{p(S_i)} \left( e^{t \left[ \mathbb{E}_{z_i^j \sim p(S_i)} \left( \ell_i(w, z_i^j) \right) - \ell_i(w, z_i^j) \right]} \right) \tag{38}
$$

$$
= e^{\frac{mnt^2 C^2}{8}}.
$$

When $t = \frac{\hat{t}}{mn}$, we have:

$$
\mathbb{E}_P \mathbb{E}_{p(S_i)} \left( e^{\frac{\hat{t}}{mn} \sum_{i=1}^{m} \sum_{j=1}^{n} \left[ \mathbb{E}_{z_i^j \sim p(S_i)} \left( \ell_i(w, z_i^j) \right) - \ell_i(w, z_i^j) \right]} \right) \leq e^{\frac{\hat{t}^2 C^2}{8mn}}, \tag{39}
$$

which can be equivalently expressed as:

$$
\mathbb{E}_P \mathbb{E}_{p(S_i)} \left( e^{\frac{\hat{t}}{m} \sum_{i=1}^{m} \left[ L_{p(S_i)}(w) - L_{S_i}(w) \right]} \right) \leq e^{\frac{\hat{t}^2 C^2}{8mn}}, \tag{40}
$$

on which we can build generalization bound of the second term. Then, we apply the Donsker and Varadhan's variational formula (Equation 2.3 on Page 13 of literature Alquier [2021]) to get:

$$
\mathbb{E}_P \mathbb{E}_{p(S_i)} \left( \sup_{\rho \in \mathcal{P}(\Theta)} e^{\frac{\hat{t}}{m} \sum_{i=1}^{m} \left[ L_{p(S_i)}(w) - L_{S_i}(w) \right] - KL(\rho \| \pi)} \right) \leq e^{\frac{\hat{t}^2 C^2}{8mn}}, \tag{41}
$$

where $\mathcal{P}(\Theta)$ denotes the set of all probability distributions and $\pi \triangleq p(w)$ is a prior distribution over hypothesis. Rearranging terms, we have:

$$
\mathbb{E}_P \mathbb{E}_{p(S_i)} \left( \sup_{\rho \in \mathcal{P}(\Theta)} e^{\frac{\hat{t}}{m} \sum_{i=1}^{m} \left[ L_{p(S_i)}(w) - L_{S_i}(w) \right] - KL(\rho \| \pi) - \frac{\hat{t}^2 C^2}{8mn}} \right) \leq 1. \tag{42}
$$

Further, replacing $\lambda$ with $\hat{t}$ and $p(w|S)$ with $\rho$, it becomes evident that:

$$
\mathbb{E}_P \mathbb{E}_{p(S_i)} \mathbb{E}_{w \sim p(w|S)} \left[ \frac{1}{m} \sum_{i \in \mathcal{M}} L_{p(S_i)}(w) - \frac{1}{m} \sum_{i \in \mathcal{M}} \left( L_{S_i}(w) + \frac{\lambda C^2}{8mn} + \frac{KL(p(w|S) \| \pi)}{\lambda} \right) \right] \leq 0. \tag{43}
$$

That is:

$$
\mathbb{E}_P \mathbb{E}_{p(S_i)} \mathbb{E}_{w \sim p(w|S)} \left[ \frac{1}{m} \sum_{i \in \mathcal{M}} \left( L_{p(S_i)}(w) - L_{S_i}(w) \right) \right] \leq \mathbb{E}_P \mathbb{E}_{p(S_i)} \mathbb{E}_{w \sim p(w|S)} \left[ \frac{\lambda C^2}{8mn} + \frac{KL(p(w|S) \| \pi)}{\lambda} \right]
$$

$$
\leq \frac{\lambda C^2}{8mn} + \frac{I(w; S)}{\lambda}. \tag{44}
$$

The choice $\lambda = \sqrt{8mnI(w; S)/C^2}$ leads to:

$$
\mathbb{E}_P \mathbb{E}_{p(S_i)} \mathbb{E}_{p(w|S)} \left[ \frac{1}{m} \sum_{i \in \mathcal{M}} \left( L_{p(S_i)}(w) - L_{S_i}(w) \right) \right] \leq \sqrt{\frac{C^2 I(w; S)}{2mn}}. \tag{45}
$$

$\square$

Then, we give the upper bound of **the first term** in Eq. (34), as follows.

This first term corresponds to the participation gap. Let $P$ be a meta distribution over $D$, where $D$ is the set of all probability distributions, i.e., $D \triangleq \{p(S_1), ..., p(S_m)\}$. With a slight abuse of notation, we define $p(w|D)$ as $\mathbb{E}_{S_i \sim p(S_i)}[p(w|S)]$. In this context, $S = \{S_1, \ldots, S_i, \ldots, S_m\}$, where $S_i$ is drawn from $p(S_i)$ and $p(S_i)$ is drawn from meta distribution $P$, we then have:

$$
\begin{aligned}
&\mathbb{E}_P \mathbb{E}_{p(S_i)} \mathbb{E}_{p(w|S)} \left[ L_{\mathcal{P}}(w) - L_{\mathcal{D}}(w) \right] \\
=& \mathbb{E}_P \mathbb{E}_{p(S_i)} \mathbb{E}_{p(w|S)} \left[ \mathbb{E}_{p(S_i) \sim P} \left[ \mathbb{E}_{z_i \sim p(S_i)}[\ell_i(w, z_i)] \right] - \frac{1}{m} \sum_{i \in \mathcal{M}} \mathbb{E}_{z_i \sim p(S_i)}[\ell_i(w, S_i)] \right] \\
=& \mathbb{E}_P \mathbb{E}_{p(w|D)} \left[ \mathbb{E}_{p(S_i) \sim P} \left[ \mathbb{E}_{z_i \sim p(S_i)}[\ell_i(w, z_i)] \right] - \frac{1}{m} \sum_{i \in \mathcal{M}} \mathbb{E}_{z_i \sim p(S_i)}[\ell_i(w, S_i)] \right].
\end{aligned}
\tag{46}
$$

For a loss function $\ell_i(w, z_i)$ bounded by an upper bound $C$, its expectation $\mathbb{E}_{z_i \sim p(S_i)}[\ell_i(w, S_i)]$ also forms a bounded loss function with an upper bound $C$. Let $\mathbb{E}_{z_i \sim p(S_i)}[\ell_i(w, S_i)]$ be a collection of independent random variables from $P$, wwe can directly derive the following conclusion based on a variant of the PAC-Bayes theorem (i.e., Theorem 2.8 in Alquier [2021]).

Let $P$ be a meta distribution over $D$, where $D$ is the set of all probability distributions, i.e., $D \triangleq \{p(S_1), ..., p(S_m)\}$. And let $\pi \triangleq p(w)$ be a prior distribution over hypothesis. Then, for any $\lambda > 0$ and for $w$ from distribution $p(w|D)$,

$$
\mathbb{E}_P \mathbb{E}_{w \sim p(w|D)}[L_{\mathcal{P}}(w)] \leq \mathbb{E}_P \mathbb{E}_{w \sim p(w|D)} \left[ L_{\mathcal{D}}(w) + \frac{\lambda C^2}{8m} + \frac{\mathrm{KL}(p(w|D) \| \pi)}{\lambda} \right],
\tag{47}
$$

where $p(w|D) = \mathbb{E}_{S_i \sim p(S_i)}[p(w|S)]$. We then have:

$$
\begin{aligned}
\mathbb{E}_P \mathbb{E}_{w \sim p(w|D)}[L_{\mathcal{P}}(w) - L_{\mathcal{D}}(w)] &\leq \mathbb{E}_P \mathbb{E}_{w \sim p(w|D)} \left[ \frac{\lambda C^2}{8m} + \frac{\mathrm{KL}(p(w|D) \| \pi)}{\lambda} \right] \\
&\leq \frac{\lambda C^2}{8m} + \frac{I(w; D)}{\lambda},
\end{aligned}
\tag{48}
$$

where $\mathbb{E}_P \left[ \mathrm{KL}(p(w|D) \| \pi) \right] = I(w; D)$. In particular, this mutual information term $I(w; D)$ cannot be "estimated", since it depends on the statistics of the "non-participating clients" which is not available. Furthermore, $I(w; D)$ is tighter than $I(w; S)$ due the convexity of the KL divergence.

The choice $\lambda = \sqrt{8mI(w; D)/C^2}$ leads to:

$$
\mathbb{E}_P \mathbb{E}_{p(S_i)} \mathbb{E}_{w \sim p(w|S)}[L_{\mathcal{P}}(w) - L_{\mathcal{D}}(w)] \leq \sqrt{\frac{C^2 I(w; D)}{2m}}.
\tag{49}
$$

Based on Eq. (49) and Eq. (35), we have:

$$
\mathbb{E}\left[ L_{\mathcal{P}}(w) - L_{\mathcal{S}}(w) \right] \leq \sqrt{\frac{C^2 I(w; S)}{2mn}} + \sqrt{\frac{C^2 I(w; D)}{2m}}.
\tag{50}
$$

$\square$

### A.4.6 Proof of Client-level Differential Privacy

Assuming that $\ell_i(w, z_i^j)$ is $L$-smooth, when $k \geq \mathcal{O}\left( \frac{\sqrt{\alpha}\epsilon^2}{\log(2/\delta)} \right)$, our proposed FedMDMI also preserves client-level $(\epsilon, \delta)$-differential privacy.

*Proof.* We have the following Gaussian mechanism for differential privacy Dwork et al. [2014]. Let $\varepsilon \in (0, 1)$ be an arbitrary number. For $c^2 > 2\ln(1.25/\delta)$, the Gaussian mechanism with standard deviation parameter $\sigma \geq 2Lc/\varepsilon$ is $(\varepsilon, \delta)$-differentially private. Wang et al. [2015] show that under this Gaussian mechanism, in standard SGLD, $k \geq \frac{\epsilon^2 n}{32\tau \log(2/\delta)}$ ensures the privacy loss to be smaller than $\frac{\epsilon\sqrt{N}}{\sqrt{32\tau k \log(2/\delta)}}$ with probability $> 1 - \frac{\tau\delta}{2nk}$, where $n$ is the total number of samples and $\tau$ represents

the samples sampled in the current iteration. We have an extra parameter $\sqrt{\alpha}$ in the noise variance. Thus, we need $k \geq \frac{\sqrt{\alpha}\epsilon^2 n}{32\tau \log(2/\delta)}$ for preserving the client-level $(\epsilon, \delta)$-differential privacy.

$\square$

