# OpenReview forum: "Improving Generalization in Federated Learning with Model-Data Mutual Information Regularization: A Posterior Inference Approach"
_NeurIPS.cc/2024/Conference — NeurIPS 2024 poster_

### Official Review · Reviewer_hP14 · 2024-07-05

**Soundness:** 3
**Presentation:** 3
**Contribution:** 3
**Rating:** 6
**Confidence:** 2

**Summary:**

This paper proposed a Federated Learning model that support Bayesian inference. To alleviate potential bias induced from local client data, a regularisation constraint on model-data mutual information is introduced. The authors show that the MCMC inference with the regularisation can be implemented through the stochastic gradient Langevin dynamics. The author also proves a generalisation bound.

**Strengths:**

- Information-theoretical modelling of federated learning is not very common. More exploration in this area is important.
- The authors provide justifications to most of the design decisions and they look sound to me.

**Weaknesses:**

- The writing is sometimes difficult to follow. The relevance of some theoretical analyses is not always clear. It would be better to introduce Algorithm 1 in the main text early on. Steps in line 10 and line 16 seem to be the key differences of the proposed method; They should be reflected in Figure 1.
- The proposed method is computationally more complex than the point estimates. The experimental results of FedMDMI do not show significantly better performance compared to FedEP. A comparison of computational cost is required to understand the potential advantages and compromises compared with all baselines.
- FedMDMI has been evaluated on small models that may not fully reflect its performance in scenarios requiring larger and more complex architectures such as ResNets. Evaluating FedMDMI on such models would provide a more comprehensive evaluation of its scalability and generalization capabilities.
- The Dirichlet distribution is used to simulate class imbalance data heterogeneity. Using heterogeneous datasets provided by TensorFlow Federated and LEAF would ensure that the method is evaluated under natural heterogeneous data distributions.

**Questions:**

- Figure 3 is very busy. Can you elaborate how they show the proposed method "ourperforms the other algorithms"? In particular, if the MI constraint regularises the effect of local client bias, why does the proposed method show volatile learning curve?
- Discuss the computational complexity of the proposed approach and how adjusting the batch size affects the performance.

---

> ### Author Rebuttal · Authors · 2024-08-07
>
> We would like to thank for the comments, and our response can be found in the following.
>
> **Comment 1-The writing is sometimes difficult to follow:**
>
> 1) Due to the space limit of the main text, we placed the theoretical proof, more detailed analysis, and Algorithm 1 in the Appendix of our submission. To improve readability, we will include these core steps of theoretical analysis and Algorithm 1 in the main text in the final version of this paper, where there is one additional page allowed.
>
> 2) Line 10 of Algorithm 1 represents the local SGLD update, which has been reflected in Figure 1 as the “SGLD for sampling” step. In the modified version, we will also reflect the global sliding average step (i.e., Line 16 of Algorithm 1) in Figure 1.
>
>
> **Comment 2-A Comparison of Computational Cost:**
>
> Please see our response to common Comment 1. Here, we first provide a complexity analysis concerning the computation time, storage, and communication for our method compared to other Bayesian methods. Subsequently, our experiments examine the relationship between computation time and model dimension within a single communication round. It is shown that the computation time of our algorithm is significantly superior to that of FedEP.
>
> **Comment 3-Additional Experiments on ResNet-18:**
>
> Following this suggestion, we have evaluated the performance of our FedMDMI by using ResNet18 on the CIFAR-10 and CIFAR-100 datasets, with the following observations.
>
> Table 13 shows that our FedMDMI still outperforms other comparison methods in terms of the generalization performance, with the ResNet-18 architecture. Here, we replace the batch norm with group norm and set the number of clients to 20, such that the 10% and 5% sampling rates correspond to only two and one client participating in training per communication round, respectively. This may highlight the robustness of our FedMDMI to some possible model-architecture changes and its ability to adapt to various models.
>
> **Comment 4-Additional  datasets provided by TensorFlow Federated and LEAF:**
>
> Please note that in the original manuscript, we used LEAF to generate the heterogeneous Shakespeare dataset, with each client associated with a speaking role comprising a few lines. Following this advice, we have further used TensorFlow Federated to generate the Stack Overflow dataset. Similar to [D1], due to the limited graphics memory and rebuttal time, we utilize only a sample of 200 clients from the original dataset, with the following observations.
>
> As shown in Table 14, on the Stack Overflow dataset, our FedMDMI continues to outperform the majority of other optimization algorithms designed to address the data heterogeneity issue. This can be attributed to our proposed model-data mutual information regularization that enhances generalization.
>
> [D1] G. Cheng _et al._, "Federated asymptotics: a model to compare federated learning algorithms," in _Proc. AISTATS_, 2023.
>
> **Comment 5-Further Clarification on Fig. 3:**
>
> i) Figures 3(a) and 3(b) provide an effective visual representation of model calibration. In a well-calibrated model, the difference between confidence and accuracy should be close to zero for each bin. Our FedMDMI (black curve) shows a closer alignment to zero across most bins, indicating superior calibration. This is further supported by Table 3, where our FedMDMI exhibits the lowest expected calibration error in most cases, particularly on the CIFAR-100 dataset.
>
> ii) Figures 3(c) and 3(d) show the convergence behavior of different algorithms. Here, we would like to show that though our FedMDMI does not converge the fastest, especially compared to the traditional optimization-based algorithm SCAFFOLD, it still slightly outperforms the other Bayesian inference-based baselines. In fact, we have also given an explanation on this observation in Line 358 of the original manuscript: "One potential explanation is our use of stochastic gradient Langevin dynamics (SGLD) to approximate the posterior, which often suffers from a slow convergence rate due to the variance introduced by the stochastic gradient."
>
> iii) Figures 3(c) and 3(d) also show that the final convergence of our FedMDMI achieves higher accuracy. This was further demonstrated in Table 2. Since the CIFAR-10 dataset is relatively simple, the baselines can easily achieve high accuracy, making our improvement less significant. However, for the more complex datasets like CIFAR-100 and Shakespeare, our FedMDMI shows a more significant improvement.
>
> The volatile learning curve mentioned by the reviewer may be due to the fact that it was plotted averaged over 5 random seeds and drawn by sampling 25 points apart, resulting in a curve that is not completely smooth. In addition, we will show here the standard deviation of the accuracy on each convergence curve when it is close to convergence (i.e., 2000-4000 rounds), as follows.
>
> For **CIFAR10 on Dir (0.2)-L**: FedAvg (0.44%), FedPA (0.74%), FedEP (0.34%), FedBE (0.51%), FALD (1.17%), FedMDMI (0.28%).
>
> For **CIFAR100 on Dir (0.2)-L**: FedAvg (0.58%), FedPA (1.19%), FedEP (0.85%), FedBE (0.69%), FALD (0.96%), FedMDMI (0.79%).
>
> Such experimental results show that, in most cases, our FedMDMI is less volatile during the convergence stage compared to other posterior inference-based algorithms.
>
> **Comment 6-Discussion on Computational Complexity:**
>
> Please also see our response to common Comment 1.
>
> For the batch size, in the original manuscript it was set to the default value of 50 for CIFAR-10 and CIFAR-100, following the same setting as in the other baselines. We have also conducted additional experiments to demonstrate the effect of batch size on performance. The results are shown in Table 12. Our method demonstrates greater robustness across different batch sizes compared to other baselines.

---

> > ### Comment · Reviewer_hP14 · 2024-08-14
> > **Thanks for the response**
> >
> > I appreciate the author's effort in improving the paper. I will raise my score.

---

> ### Author Response · Authors · 2024-08-14
> **Official Comment by Authors:**
>
> Dear Reviewer hP14,
>
> Many thanks for the time and effort that you have dedicated to reviewing our paper and providing these insightful comments, which will further help enhance the quality of the final version of our manuscript.

---

### Official Review · Reviewer_XkDq · 2024-07-09

**Soundness:** 2
**Presentation:** 3
**Contribution:** 2
**Rating:** 7
**Confidence:** 3

**Summary:**

The paper proposes an approach to mitigate training failure in the heterogeneous federated learning setup. The approach combines Bayesian perspective of posterior inference on the client side and regularization of mutual information between weights and data in order to reduce the effect of difference in the local datasets. The authors provide computable approximations of the values involved and provide information theoretic bound on the generalization in federated setup. Experiments show that the method succeeds and outperforms several baselines.

**Strengths:**

The paper gives a detailed explanation of the approach for heterogeneous federated learning with mutual information regularization and posterior estimation. A generalization bound is provided and extensive experiments are performed.

**Weaknesses:**

The motivation to introduce posterior inference in federated setup is not clear: The problem of the point estimation and inability to have uncertainty of the predictions is equally valid for the centralized setup as well. The further description of the approach is convoluted, it seems that the main reason to resort to posterior estimation is to be able to obtain tractable computation for the mutual information term and corresponding generalization bound from PAC-Bayes perspective.

The method is motivated by scarce local data that might lead to overfitting and prevent from training a global model when aggregated, but this setup is not checked empirically. Only the heterogeneity with respect to labels distribution is evaluated.

**Questions:**

1 - In equation (20) there are two different mutual information terms, I(w;S) and I(w;D). What is the difference between them?

2 - What are the conditions under which it is possible to decompose posterior of the global model into the product of posteriors of the local models?

3 - In the conclusion you claim to show that the optimal posterior is Gibbs posterior, but as I understand you used this result from previous research?

**Limitations:**

Limitations are addressed.

---

> ### Author Rebuttal · Authors · 2024-08-07
>
> We would like to thank for the comments, and our response can be found in the following.
>
> **Comment 1-Further clarification on Our Motivation:**
>
> We agree with the reviewer that in centralized learning, posterior inference is proposed to provide a more reliable assessment of model uncertainty than point estimation. This uncertainty estimation is also crucial in the safety-critical applications of federated learning, such as autonomous driving, healthcare, and finance. Consequently, there has been some recent literature, such as FedPA and FedEP, considering the introduction of posterior inference into the federated setup.
>
> However, we found that these Bayesian inference-based federated learning algorithms do not adequately address the data heterogeneity issue. Therefore, we are motivated to propose a mutual information regularization to enforce the global posterior to learn essential information from the heterogeneous local data, thereby improving the generalization capability. To optimize this regularization, we further employ a series of techniques, including the global mutual information decomposition, PAC-Bayesian conclusions, and stochastic gradient Langevin dynamics (SGLD) sampling.
>
> **Comment 2-Additional Experiments on Scarce Local Data:**
>
> Thank you for your comments. We have conducted additional experiments to analyze the impact of the number of local data samples. Given that the total number of data samples in CIFAR-10 and CIFAR-100 is fixed, we controlled the number of samples on each client by adjusting the number of clients. With fewer data samples on a client, the local model is more prone to overfitting. The results are shown in Table 11. Our FedMDMI maintains superior performance even with scarce local data, demonstrating that our MI regularization-based posterior estimation effectively alleviates the overfitting caused by data scarcity.
>
> **Comment 3-The difference between mutual information terms, I(w; S) and I(w; D):**
>
> In Eq. (20), $I(w; S)$ relates to the participating generalization error, i.e., the difference between the empirical and expected risk for the participating clients. This serves as the regularized term that can be estimated in our FedMDMI. On the other hand, $I(w; D)$ relates to the participation gap, i.e., the difference in the expected risk between the participating and non-participating clients. This term, however, cannot be estimated due to the unavailability of the non-participating clients.
>
> Here, we will illustrate the difference with a simple example. Consider an extreme case with 10 clients, each holding only one label of the MNIST dataset. In this scenario, we restrict federated training on the first 9 clients, which contain train dataset labeled 0-8. During the test phase, the error measured on test sets from labels 0-8 is referred to as the participating generalization error $I(w; S)$. The error measured on the data from the $10$-th client, which holds label 9, is termed as the participation gap  $I(w; D)$.
>
> Thus, $I(w;D)$ and $I(w;S)$ do not represent a compromise or antagonistic relationship. By optimizing $I(w;S)$, we can improve the performance of the learned model on the distribution seen during training. Consequently, if the distribution of clients that have never participated is the same as the distribution seen by trained clients, then $I(w;D)$ degenerates to $I(w;S)$. In other words, our FedMDMI can only guarantee improved generalization on the distribution seen during training, but not on the out-of-distribution data from non-participating clients. This could motivate an interesting future direction: reducing $I(w;D)$ through techniques like transfer learning or domain generalization.
>
> **Comment 4-What are the conditions under which it is possible to decompose the posterior of the global model into the product of posteriors of the local models?**
>
> The decomposition of the global posterior into the product of posteriors of the local models is based on two assumptions:
>
> 1. The global likelihood is conditionally independent given $w$, i.e., $p(S_1,\dots,S_m | w) = \prod_{i=1}^m p(S_i | w)$.
> 2. The ratio of the global prior to the product of client priors, $\frac{p(w)}{\prod_{i=1}^m p_i(w)}$, is considered a constant based on some prior assumptions, such as Gaussian priors or uniform priors.
>
> These assumptions are widely recognized and significant across various fields, including in PoE [C1] and (EP)-MCMC [C2, C3], and Bayesian FL [C4].
>
> [C1] Hinton, Geoffrey E., "Training products of experts by minimizing contrastive divergence," Neural computation 14(8): 1771-1800, 2002.
>
> [C2] Neiswanger, Willie, Chong Wang, and Eric P. Xing, "Asymptotically exact, embarrassingly parallel MCMC," in _Proc. UAI_, 2014.
>
> [C3] Wang, Xiangyu, and David B. Dunson, "Parallelizing MCMC via Weierstrass sampler," arXiv preprint arXiv:1312.4605, 2013.
>
> [C4] Al-Shedivat, Maruan, _et al._, "Federated learning via posterior averaging: A new perspective and practical algorithms," in _Proc. ICLR_, 2021.
>
> **Comment 5-In the conclusion, you claim to show that the optimal posterior is the Gibbs posterior, but as I understand, you used this result from previous research?**
>
> We agree with the reviewer that this is not our contribution. To avoid confusion, we emphasized in the original manuscript that "this conclusion is well-established in the field of PAC-Bayesian learning" in the Introduction (Line 64). We also stated that, "In order for our paper to be self-contained, we re-state the proof from [12, 44] here for the optimal posterior" in the Proof (Line 988). To avoid overclaiming our contribution, we will further emphasize that the result of the optimal posterior being the Gibbs posterior stems from previous research.

---

> > ### Comment · Reviewer_XkDq · 2024-08-09
> >
> > I thank the authors for the rebuttal and raise my score.

---

> > > ### Author Response · Authors · 2024-08-13
> > > **Official Comment by Authors:**
> > >
> > > Dear Reviewer XkDq,
> > >
> > > We sincerely appreciate the time and effort that you have dedicated in reviewing our paper and providing these insightful comments, which will further help improve the quality of the final version of our manuscript.

---

### Official Review · Reviewer_re5T · 2024-07-10

**Soundness:** 3
**Presentation:** 3
**Contribution:** 3
**Rating:** 6
**Confidence:** 4

**Summary:**

In this paper, the authors introduce a method for federated learning to bypass problems caused by inter-client data heterogeneity.
For this, they introduce a Bayesian approach with information-theoretic regularizer, that will prevent local models from overfitting. Specifically, the authors add a model-data regularizer at a global level and then show how it could be computed in a federated fashion. The local optimal posterior appeared to be the Gibbs posterior, and the authors employ SGLD to sample from it. To show the efficacy of the approach, authors conduct a series of experiments on image and text data at federated datasets.

**Strengths:**

I think the paper has the following strengths:
- I find the paper very easy to follow, and the suggested idea is very interesting and natural;
- It tackles an important problem of data heterogeneity in Federated Learning;
- The paper provides detailed theoretical derivations and experimental evaluation;

**Weaknesses:**

I find the following things are downsides:
- The loss used in the optimization is a result of several levels of approximations.
First, the global model-data MI term is upper-bounded by the sum of local model-data MI terms.
Second, each local model-data term is itself upper-bounded by the RHS of Eq. 17.
I think that the discussion on the tightness of the upper bound is missing.

- I feel that some baselines are missing. Probably the first and classical approach to combat the problem of data heterogeneity was FedProx [1] (which was cited), but it is not compared with.
Also, it would be interesting to see, where the method is placed if compared with personalized approaches, that are specially built to deal with data heterogeneity. E.g. FedPop [2] (Bayesian), FedRep [3] (not Bayesian).

[1] Li T. et al. Federated optimization in heterogeneous networks //Proceedings of Machine learning and systems. – 2020. – Т. 2. – С. 429-450.

[2] Kotelevskii N. et al. Fedpop: A bayesian approach for personalised federated learning //Advances in Neural Information Processing Systems. – 2022. – Т. 35. – С. 8687-8701.

[3] Collins L. et al. Exploiting shared representations for personalized federated learning //International conference on machine learning. – PMLR, 2021. – С. 2089-2099.

- Minor typos:
1) Lines 142-143 collapsed (negative vspace?);
2) Figure 2: The left y-axis lives in [0, 1], right y-axis in [0, 100].

**Questions:**

- In Line 76 it is said that the MI term (...) "offers a certain client-level privacy protection as a byproduct."
I think the authors imply that the MI term regularizes the fitness of a local model. I wonder, how overfitness can compromise the privacy of training data.
- In lines 171-175 there is a reasoning about some bias factor $\delta$ that affects the generation of local data. What is $\delta$? Why does inequality in 174 hold? Can you elaborate more on the last sentence in lines 174-175?
- In lines 212-213 authors use the term Variational Approximation (VA). I have a feeling that the term Variational Inference (VI) is a more common alternative to MCMC met in literature.

**Limitations:**

The authors have a separate section on limitations. However, not all of them are addressed (see the Weaknesses section, about upper-bound on a loss).

---

> ### Author Rebuttal · Authors · 2024-08-07
>
> We would like to thank for the comments, and our response can be found in the following.
>
> **Comment 1-Discussion on tightness of the upper bound:**
>
> First, we are constrained in practice to only leverage local data $S_i$ at individual client $i$ under the FL settings. Alternatively, based on the Global Model-Data MI Decomposition in Proposition 4.1, we have: $$I(w ; S) = \sum_{i=1}^m \left[ I\left(w ; S_i\right) - I\left(S_i ; S^{i-1}\right) \right]
> \le \sum_{i=1}^m I\left(w ; S_i\right).$$
> Here, we can upper bound $I(w ; S)$ by the sum of local MI terms $I(w ; S_i)$ that can be estimated locally by these clients. By doing so, we also introduce an estimation error, i.e., the data correlation $I\left(S_i ; S^{i-1}\right)$ between the individual clients, which determines the tightness of the proposed upper bound $\sum_{i=1}^m I\left(w ; S_i\right)$. This data correlation $I\left(S_i ; S^{i-1}\right)$ between the individual clients is related to the clients' data generation and collection, which is intractable to estimate and optimize.
>
> Then, each local MI term $I(w; S_i)$ can be further expressed as:
> $$ I(w ; S_i) = \mathbb{E}\_{p(S_i)}[\operatorname{KL}(p(w \mid S_i) \| p_i(w))]
>   = \mathbb{E}\_{p(S_i)}[\operatorname{KL}(p(w \mid S_i) \| r(w))]-\operatorname{KL}[p_i(w) \| r(w)] \leq \mathbb{E}\_{p(S_i)}[\operatorname{KL}(p(w \mid S_i) \| r(w))]\],$$
> where $p_i(w) \triangleq \mathbb{E}_{p(S_i)}[p(w | S_i)]$ denotes the oracle prior and  $r(w)$ is an arbitrary prior distribution that is used to approximate $p_i(w)$. This may incur another estimation error, which is the difference between the oracle prior $p_i(w)$ and the prior $r(w)$ we actually use, i.e., $\operatorname{KL}\left[p_i(w) \| r(w)\right]$.
>
> As indicated in Line 240, [B1] emphasizes that to achieve a small KL divergence, the prior must, in essence, predict the posterior. Specifically, [B1] employs a distribution of pre-trained models derived from a portion of the untrained data as prior $r(w)$.
>
> In our work, we propose using the global model in the previous round as the mean $\mu$ of Gaussian prior and the uncertainty introduced by all clients in the previous round as the covariance $\Sigma^{-1}$ in the prior, as shown in Eq. (18) and Line 6 in Algorithm 1 of our FedMDMI. This incorporates the global data information and potentially helps predict the local priors based on the global posterior decomposition.
>
> [B1] G. Dziugaite _et al._, "On the role of data in PAC-Bayes bounds," in _Proc. ICML_, 2021.
>
> **Comment 2-Additional baselines:**
>
> Please see our response to common Comment 2.
>
> **Comment 3-Minor typos:**
>
>  Thank you for pointing out the small issue with collapsed Lines 142-143, and we will fix it. Additionally, in Figure 2, the ordinate represents the percentage error. Specifically, the left figure indicates that the training error is within the range of [0, 1%], while the right figure indicates that the test error is within the range of [0, 20%].
>
> **Comment 4-Relationship between Overfitness and Privacy of Training Data:**
>
> The MI term $I(w;S)$ regularizes the fitness of a local model. In fact, the hyperparameter $\alpha$ controls the degree of the MI regularization and embodies a trade-off between privacy and fitting (or trade-off between generalization and fitting). As highlighted in Line 275, the proposed MI regularizer $I(w;S)$ directly quantifies the extent to which the model memorizes data. A larger value of $I(w;S)$ implies that the model fits the data well, thus signifying the fitting. Conversely, a smaller $I(w;S)$ indicates that the model avoids memorizing excessive data details, thereby promoting generalization and, from another perspective, privacy protection.
>
> In fact, some work [B2] demonstrates that in certain cases, knowing the previous model and the gradient update from a client can allow one to infer a training example held by that user. Therefore, for stronger privacy protection, differential privacy can be used to encrypt these models or gradient updates. Our work shows that regularizing the MI term can provide a form of differential privacy protection as a byproduct.
>
> [B2] P. Kairouz _et al._, Advances and open problems in federated learning, Foundations and Trends in machine learning, 2021.
>
> **Comment 5- Further Clarification on Factor $\delta$ :**
>
> Here, we would like to explore and offer further insights, from the perspective of FL, on how minimizing mutual information can help mitigate bias resulted from the data heterogeneity.
>
> In FL, we denote the bias factor as $\delta$ that affects the generation of local heterogeneous data (such as the difference in clients' locations, preferences, or habits). If we can effectively eliminate the influence of the bias factor $\delta$, the entire dataset would become homogeneous (similar), thereby reducing bias among local posteriors.
>
> Indeed, directly eliminating the impact of bias factor is impractical. But leveraging Markov chain $\delta \rightarrow S \rightarrow w$ allows us to directly infer that $I(w; \delta) \le I(w; S)$, where the inequality holds due to the **Data Processing Inequality (DPI)**. As a consequence, we can diminish $I(w; S)$ to constrain $I(w; \delta)$, thereby rendering the model $w$ insensitive to the bias factor $\delta$ from the diverse clients.
>
> Furthermore, similar analyses are prevalent in other contexts. For instance, in centered learning, literature [B3] (Section 2.2 on Page 6) assumes that nuisances affect the observed data, and these nuisances are mitigated through information bottleneck regularization (another well-known mutual information).
>
> [B3] A. Achille _et al._, "Emergence of invariance and disentanglement in deep representations," Journal of Machine Learning Research, 2018.
>
>
> **Comment 6-Comparison between MCMC and Variational Approximation (VA):**
>
> Please see our response to common Comment 3.

---

> > ### Comment · Reviewer_re5T · 2024-08-12
> >
> > Dear authors,
> >
> > Thank you for your detailed response.
> >
> > I appreciate how you addressed the concerns I raised. Therefore, I am willing to raise my score from 5 to 6.
> >
> > I suggest adding results from new experiments to the revised version, including the running time, to make your evaluation more complete.
> >
> > It would also be helpful to mention relevant literature on Bayesian Federated Learning to give readers more context.

---

> > > ### Author Response · Authors · 2024-08-13
> > > **Official Comment by Authors:**
> > >
> > > Dear Reviewer re5T,
> > >
> > > We sincerely appreciate your time and effort in reviewing our paper and providing valuable comments. We will incorporate these additional baselines and new supplementary experiments into the revised version. Additionally, personalized Bayesian federated learning will also be included in the discussion. These insights will further help enhance the quality of the final version of our manuscript.

---

### Official Review · Reviewer_4nGu · 2024-07-10

**Soundness:** 3
**Presentation:** 3
**Contribution:** 3
**Rating:** 6
**Confidence:** 2

**Summary:**

The paper considers the problem of Bayesian Federated Learning when there’s data heterogeneity and class imbalance across clients and develops a posterior inference approach for model parameters through mutual information regularization of the data and global parameters in local posteriors. This is achieved via using the KL formulation of the mutual information and showing that the optimal local posterior is a Gibbs distribution. To infer local posterior Stochastic Gradient Langevin Dynamics is used and to aggregate the local posteriors into the global posteriors a simple average of the local samples is considered. The theoretical results suggest that the proposed algorithm results in the convergence of the local and global posteriors and corresponding generalization bounds are provided. Results are shown on a few datasets where competitive test performance and uncertainty calibration are achieved.

**Strengths:**

*   The problem considered is timely and important. In almost every application of federated learning, client data heterogeneity and scarcity are inevitable. For data scarcity, considering the model uncertainty (through Bayesian modeling) and for data heterogeneity removing the biases of local and global posteriors due to biases in local datasets are sensible directions.

*   Accompanying the empirical results and intuitions with theoretical arguments is another strength of the paper.

*   The writing is clear and understandable, and the organization of the arguments is intuitive and logical.

**Weaknesses:**

*   Although competing (Bayesian) methods are not specifically designed to deal with data heterogeneity there seems to be very little to no gain achieved by the model. The results in the current form seem weak to me. The performance gains are quite marginal (e.g. Fig. 3) both for uncertainty calibration and test performance. Are the performance differences provided in tables 2, and 3 statistically significant? Shouldn’t we assume that as the degree of heterogeneity increases the proposed method is more effective? Why don’t we see a larger gap as a function of $\\alpha$?

*   A few important papers seem to be missing from the introduction such as \[1\]. Some methods are not considered for the comparisons such as \[2,3\].

*   The only information about the time (and memory) complexity is a sentence somewhere close to the end of the paper. While an important motivation of the paper is to reduce the computational and memory cost, empirical timing results aren’t shown. Can you include a detailed timing comparison as a function of data heterogeneity, dimension and size of the network, etc? Intuitively variational methods should do a much better job compared to sampling-based methods. Although the stochastic version of the Langevin Dynamics is used here still they should take much longer than variational methods to converge.


\[1\] Cao, Longbing, et al. "Bayesian federated learning: A survey." arXiv preprint arXiv:2304.13267 (2023).

\[2\] Chen, Hui, et al. "FedSI: Federated Subnetwork Inference for Efficient Uncertainty Quantification." arXiv preprint arXiv:2404.15657 (2024).

\[3\] Kim, Minyoung, and Timothy Hospedales. "Fedhb: Hierarchical Bayesian federated learning." arXiv preprint arXiv:2305.04979 (2023).

**Questions:**

Even though I went through the paper in full I’m still not fully convinced that mutual information regularization is a good idea. The bound provided in Eq. 20 suggests that the generalization error is bounded by two mutual information terms one of which is the target of estimation of the paper. How can we argue that there’s no tradeoff between lowering the first and second terms? Why does reducing the first term result in a lower generalization error?

**Limitations:**

See the weaknesses section.

---

> ### Author Rebuttal · Authors · 2024-08-03
>
> We would like to thank for the comments, and our response is as follows.
>
> **Comment 1-Further Clarification on Fig.3:** Fig. 3 visualizes uncertainty calibration and convergence, with quantitative results in Tables 2 and 3 highlighting improvements of our FedMDMI.
>
> i) Figs. 3(a) and 3(b) provide an effective visual representation of model calibration. In a well-calibrated model, the difference between confidence and accuracy should be close to zero for each bin. Our FedMDMI (black curve) shows a closer alignment to zero across most bins, indicating a superior calibration. This is further supported by Table 3, where our FedMDMI exhibits the lowest expected calibration error in most cases, particularly on CIFAR-100 dataset.
>
> ii) Figs. 3(c) and (d) illustrate convergence behavior. Although FedMDMI converges slower than SCAFFOLD, it slightly outperforms other Bayesian baselines. This is explained in Line 358 of the manuscript, attributing the slower convergence to the variance from stochastic gradient Langevin dynamics (SGLD).
>
> iii) Figs. 3(c) and (d)  also show that the final convergence of our FedMDMI achieves higher accuracy. This was further demonstrated in Table 2. Since CIFAR-10 dataset is relatively simple, the baselines can easily achieve high accuracy, making our improvement less significant. However, for the more complex datasets like CIFAR-100 and Shakespeare, our FedMDMI shows a more significant improvement.
>
> iv) Regarding our response to Comment 6, the computational and storage complexity of our FedMDMI is lower than that of most other Bayesian FL algorithms.
>
> **Comment 2-Performance differences in Tables 2 and 3:** Please note that results presented in Tables 2 and 3 are averaged over five random seeds, as indicated in the caption of both tables.
>
> **Comment 3- Degree of heterogeneity and effectiveness:** Yes, we agree with the reviewer. As the degree of heterogeneity increases, performance of all baselines decreases, but our method's performance decreases less significantly. This indicates that our FedMDMI is more robust to increasing heterogeneity compared to the baseline.
>
> Specifically, for CIFAR-10, as the degree of heterogeneity increases from Dir(0.7)-H to Dir(0.2)-H, performance of our FedMDMI and FedAvg decreases by 0.2% and 0.63%, respectively. Similarly, for CIFAR-100, performance of our FedMDMI and FedAvg decreases by 1.01% and 2.03%, respectively. This trend holds in most cases, indicating that our proposed posterior inference approach based on model-data mutual information regularization can effectively alleviate the impact of data heterogeneity.
>
> **Comment 4-Discussion of $\alpha$:** The Lagrange multiplier $\alpha$ in Eq. (7) balances fitting ($L_{S_i}(w)$) and generalization ($I(w; S_i)$). Fig. 2 illustrates that as $\alpha$ increases, the gap between test errors across varying data heterogeneity decreases, indicating the efficacy of our proposed regularization in addressing data heterogeneity. Specifically, increasing $\alpha$ leads to a gradual rise in train error and an initial drop followed by an increase in test error, suggesting that lower complexity (larger $\alpha$) may cause underfitting, while higher complexity (smaller $\alpha$) risks overfitting. Thus, larger $\alpha$ mitigates the bias from data heterogeneity.
>
> **Comment 5-Additional Baselines:**  Please see common comment 2.
>
> **Comment 6-Complexity Analysis:**  Please see common comment 1.
>
> **Comment 7-Comparison of MCMC and VA:** Please see common comment 3.
>
> **Comment 8-Effectiveness of MI regularization:**  As discussed in the Related Work section, Model-Data Mutual Information (MDMI) regularization is a well-established concept in centralized learning, supported by both theoretical analyses and extensive experimental verification. For instance, [A1] demonstrates that MDMI regularization outperforms L2-norm regularization and dropout through Fisher information matrix estimation.
>
> For FL, we propose a federated posterior inference method to estimate the global mutual information via global MDMI Decomposition. Our experiments indicate that MDMI regularization is equally effective in FL.
>
> [A1] Z. Wang *et al.*, "Pac-Bayes information bottleneck." in *Proc. ICLR*, 2022.
>
> **Comment 9-Difference between mutual information terms:**
> In Eq. (20), $I(w; S)$ relates to the participating generalization error, i.e., the difference between the empirical and expected risk for participating clients. This serves as the regularized term that can be estimated in our FedMDMI. On the other hand, $I(w; D)$ relates to the participation gap, i.e., difference in the expected risk between participating and non-participating clients. This term, however, cannot be estimated due to unavailability of the non-participating clients.
>
> Here, we will illustrate the difference with a simple example. Consider an extreme case with 10 clients, each holding only one label of MNIST dataset. In this case, we restrict federated training on the first 9 clients, which contain train dataset labeled 0-8. During the test phase, the error measured on test sets from labels 0-8 is referred to as the participating generalization error $I(w; S)$. The error measured on data from the $10$-th client, which holds label 9, is termed as the participation gap $I(w; D)$.
>
> Thus, $I(w;D)$ and $I(w;S)$ do not represent a compromise or antagonistic relationship. By optimizing $I(w;S)$, we can improve performance of the learned model on the distribution seen during training. Consequently, if the distribution of clients that have never participated is the same as the distribution seen by trained clients, then $I(w;D)$ degenerates to $I(w;S)$. In other words, our FedMDMI can only guarantee improved generalization on the distribution seen during training, but not on the out-of-distribution data from non-participating clients. This could motivate an interesting future direction: reducing $I(w;D)$ through techniques like transfer learning or domain generalization.

---

> > ### Comment · Reviewer_4nGu · 2024-08-12
> > **Updated review**
> >
> > Thank you for your detailed explanation of the figures and results. The new experiments and results are helpful for better understanding the empirical aspects of the contribution. Please see below for an updated review.
> >
> > **Comment 1-Further Clarification on Fig.3:** I understand what the plots are representing. To me, it’s surprising that the lines all lie very close together and very far from zero (the optimal line). Adding shaded error bars representing variance to all plots in Fig. 3 would be very helpful.
> >
> > **Comment 2-Performance differences in Table 3:** Can you report the variance around those mean performances as well?
> >
> > **Comment 3- Degree of heterogeneity and effectiveness:** The pattern you’re referring to is not consistent across all methods (e.g. FedEP). Since no other model is designed to handle data heterogeneity I’m very surprised that FedMDMI doesn’t outperform all other models by a large margin as the heterogeneity hyperparameter increases. Can you make a line plot of the ECE and top-1 accuracy as a function of heterogeneity to see if the line corresponding to FedMDMI diverges at some point?
> >
> > **Comment 1-Complexity analysis:** As I understand, the computational and memory complexity is provided for a single round of SGLD. The main question is how long it takes for each method to converge. Depending on the problem this could result in better or worse convergence than optimization-based methods [1,2]. In general, it’s believed that for real-world problems variational methods are faster in high dimensions than sampling-based methods. Therefore I’m surprised that the timing results in the rebuttal are in favor of FedMDMI. Do the authors have any explanation for this?
> >
> > **Comment 3-Comparison of MCMC and Variational Approximation (VA):** While the motivation is solid and sensible, ultimately it’s an empirical question whether variational methods or sampling-based methods perform best for FL with data heterogeneity. Is any of the compared methods based on a variational framework?
> >
> > [1] Ma, Yi-An, et al. "Sampling can be faster than optimization." Proceedings of the National Academy of Sciences 116.42 (2019): 20881-20885.
> >
> > [2] Kim, Kyurae, Yian Ma, and Jacob Gardner. "Linear Convergence of Black-Box Variational Inference: Should We Stick the Landing?." International Conference on Artificial Intelligence and Statistics. PMLR, 2024.

---

> ### Author Response · Authors · 2024-08-13
> **Official Comment by Authors:**
>
> Dear Reviewer 4nGu,
>
> Thank you for the updated review. We have provided our response to each of these concerns in the following.
>
> **Comment 1-Further Clarification on Fig. 3:** Regarding the plots in Figs. 3(a) and 3(b), as noted by the reviewer, the lines all lie very close together and very far from zero. This observation actually highlights a key challenge in the FL settings: the scarcity of data samples at each client increases the risk of local model overfitting, leading to overconfident decisions and poor uncertainty estimation in the aggregated global model. This also underscores the necessity of incorporating Bayesian inference into FL. As further illustrated in Table 3, the posterior-based approach demonstrates a significantly better performance in uncertainty estimation as compared to the point-based approach.
>
> Besides, following this suggestion, we will also include the shaded error bars representing variance in all plots of Fig. 3 in the final version of our manuscript.
>
> **Comment 2-Performance differences in Table 3:** Following this suggestion, we will include the variance of the ECE values in Table 3, as follows.
>
> **Table 1: ECE (with  the variance of the ECE values) under various setting.**
> | Method   | **CIFAR-10** |  | **CIFAR-10** | |
> |----------|-------------|-------------|-------------|-------------|
> | |Dir (0.2)-L | Dir (0.7)-L | Dir (0.2)-L | Dir (0.7)-L  |
> | FedAvg   | 0.169 ± 0.0039 | 0.165 ± 0.0042 | 0.429 ± 0.0030 | 0.432 ± 0.0028 |
> | FedM     | 0.159 ± 0.0025 | 0.169 ± 0.0036 | 0.468 ± 0.0027 | 0.459 ± 0.0035 |
> | MimeLite | 0.182 ± 0.0041 | 0.178 ± 0.0034 | 0.461 ± 0.0029 | 0.470 ± 0.0022 |
> | SCAFFOLD | 0.192 ± 0.0018 | 0.194 ± 0.0025 | 0.472 ± 0.0016 | 0.479 ± 0.0034 |
> | FedBE    | 0.182 ± 0.0029 | 0.189 ± 0.0032 | 0.440 ± 0.0015 | 0.463 ± 0.0021 |
> | FedPA    | 0.173 ± 0.0031 | 0.176 ± 0.0020 | 0.374 ± 0.0033 | 0.371 ± 0.0025 |
> | FedEP    | 0.121 ± 0.0033 | **0.118 ± 0.0021** | 0.289 ± 0.0045 | 0.273 ± 0.0027 |
> | FALD     | 0.135 ± 0.0028 | 0.127 ± 0.0023 | 0.267 ± 0.0018 | 0.269 ± 0.0023 |
> | FedMDMI  | **0.115 ± 0.0019** | 0.120 ± 0.0031 | **0.261 ± 0.0023** | **0.263 ± 0.0029** |
>
> **Comment 3- Degree of heterogeneity and effectiveness:**
> First, we would like to clarify that most of the Bayesian-based comparison methods, including FedBE, FedPA, and FedEP, are specifically designed to address the data heterogeneity issue in FL. We apologize for not clarifying this point earlier due to the word limit on the first-round response.
>
> Besides, following the reviewer's suggestion, we have also conducted additional experiments (iid and Dir (0.1)), observing that as the degree of heterogeneity increases, the generalization performance of all the comparison algorithms declines. However, the generalization performance of our FedMDMI decreases to a lesser extent in the most cases. This indicates that our FedMDMI is more robust to the increasing heterogeneity as compared to the other baselines. Since we cannot display images here, we present these results in the following table form for now, and we will convert it to line plots in the final version of our manuscript.
>
> We have not yet reached a conclusion on how data heterogeneity affects the ECE. It is clear that uncertainty estimation is influenced by local sample size, and smaller sample sizes tend to lead to overconfident decisions. On the other hand, data heterogeneity has a more pronounced effect on accuracy.
>
> **Table 2: Test Accuracy (\%) v.s. Data heterogeneity.**
> | Method  | **CIFAR10** |   |   |   | **CIFAR100** |   |  |  |
> |---------|----------------|----------------------|----------------------|----------------------|-----------------|-----------------------|-----------------------|-----------------------|
> |  | iid-H (CIFAR10) | Dir (0.7)-H (CIFAR10) | Dir (0.2)-H (CIFAR10) | Dir (0.1)-H (CIFAR10) | iid-H (CIFAR100) | Dir (0.7)-H (CIFAR100) | Dir (0.2)-H (CIFAR100) | Dir (0.1)-H (CIFAR100) |
> | FedAvg  | 83.22           | 80.31                | 79.68                | 77.96                | 48.65           | 42.35                 | 40.32                 | 38.14                 |
> | FedBE   | 84.06           | 82.33                | 81.25                | 79.19                | 51.68           | 46.29                 | 44.82                 | 42.06                 |
> | FedPA   | 84.82           | 82.93                | 82.78                | 80.37                | 52.44           | 49.66                 | 48.51                 | 46.79                 |
> | FedEP   | 84.93           | **83.79**            | 83.30                | 81.23                | 52.99           | 50.02                 | 49.08                 | 47.32                 |
> | FedMDMI | **85.05**       | 83.76                | **83.56**            | **82.28**            | **53.28**       | **50.71**             | **49.70**             | **48.41**             |

---

> ### Author Response · Authors · 2024-08-13
> **Official Comment by Authors:**
>
> **Table 3: ECE v.s. Data heterogeneity.**
> | Method  | **CIFAR10** |   |   |   | **CIFAR100** |   |  |  |
> |---------|----------------|----------------------|----------------------|----------------------|-----------------|-----------------------|-----------------------|-----------------------|
> |  | iid-H (CIFAR10) | Dir (0.7)-H (CIFAR10) | Dir (0.2)-H (CIFAR10) | Dir (0.1)-H (CIFAR10) | iid-H (CIFAR100) | Dir (0.7)-H (CIFAR100) | Dir (0.2)-H (CIFAR100) | Dir (0.1)-H (CIFAR100) |
> | FedAvg  | 0.174           | 0.170                | 0.168                | 0.160                | 0.437           | 0.440                 | 0.434                 | 0.430                 |
> | FedBE   | 0.181           | 0.184                | 0.187                | 0.169                | 0.452           | 0.467                 | 0.438                 | 0.442                 |
> | FedPA   | 0.179           | 0.174                | 0.167                | 0.168                | 0.387           | 0.380                 | 0.382                 | 0.361                 |
> | FedEP   | 0.137           | 0.124                | 0.130                | 0.125                | 0.281           | 0.284                 | 0.298                 | 0.277                 |
> | FedMDMI | **0.133**       | **0.122**            | **0.125**            | **0.117**            | **0.272**       | **0.262**             | **0.267**             | **0.259**             |
>
> **Comment 1-Complexity analysis:** We did provide the complexity analysis in terms of the computation and memory for a single communication round. To calculate the total time consumption required to reach the convergence, we then have to multiply the computation time per round with the total number of communication rounds $T$ needed for convergence. In the following, we begin by providing a theoretical analysis of the total number of communication rounds required for convergence.
>
> **_i) Theoretical convergence analysis._** For our proposed **FedMDMI**, as shown in Lines 230-232 of our manuscript, and leveraging the insights from [A3], we provide the convergence rate of our **FedMDMI** under the $\mu$-strongly convex objective as: $\mathcal{O}\left((1-\gamma \mu /8)^{KT} + {1/m} + d\right)$, where $m$ is the number of clients, $K$ is the number of local updates, $d$ is dimensions of the model, and $T$ is the number of communication rounds. Thus, the convergence rate scales linearly with the model dimension $d$. Reference [A4] shows that the convergence rate of **FedAvg** on the $\mu$-strongly convex objective is: $\mathcal{O}\left(\mu\exp({-\frac{\mu}{16(1+B^2)}T}) +{1/(mKT)}\right)$, where $B$ is the gradient dissimilarity. Notably, neither **FedPA** nor **FedEP** presented the convergence rate in their original papers.
>
> **_ii) Empirical convergence behavior._** From the convergence curves illustrated in our manuscript (i.e., Figs. 3(c) and 3(d)), it can be empirically observed that the number of communication rounds required for convergence of our **FedMDMI** is less than that of **FedPA** and **FedEP**.
>
> Then, we try to respond to the reviewer's concern on variational approximation (VA) methods being generally faster than sampling-based approaches in real-world problems. It is however observed in our rebuttal that FedEP, which uses the expectation propagation to obtain a VA of the global posterior, consumes more time per communication round and requires a greater number of communication rounds compared to our FedMDMI. The reasons are explained in the following.
>
> **_i) Complexity comparison per communication round._** As stated in our first-round response to the common comment 1, **FedMDMI** only requires generating and computing the Gaussian noise at each round, while the sequence of global model $w_t$ converges to Gibbs posterior with sufficiently large $t$. In contrast, **FedEP** requires approximating the covariance as the inverse Hessian at each round, which introduces an additional $\mathcal{O}(d^3)$ time complexity and $\mathcal{O}(d^2)$ memory complexity.
>
> **_ii) Total communication rounds._** In fact, convergence analysis for the federated variational approximation is still an open question, with no established results at present. So we try to give an intuitive explanation on why VA may converge slower than MCMC in FL. While VA may outperform sampling-based methods in terms of the convergence in centralized learning or real-world cases, the situation is different in FL. The main drawback of VA is that it typically results in biased posterior estimates for complex posterior distributions. This issue is further exacerbated in FL, where data heterogeneity across clients already contributes to biased local posteriors, making the use of VA even more challenging. Consequently, using VA in this context can result in an unstable and slow convergence of the aggregated global posterior.

---

> > ### Author Response · Authors · 2024-08-13
> > **Official Comment by Authors:**
> >
> > [A3] Plassier _et al._, "Federated averaging langevin dynamics: Toward a unified theory and new algorithms," in _Proc. AISTATS_, 2023.
> >
> > [A4] Karimireddy, Sai Praneeth _et al._, "SCAFFOLD: Stochastic controlled averaging for federated learning," in _Proc. ICML_, 2020.
> >
> > **Comment 3-Comparison of MCMC and Variational Approximation (VA):** Within the comparison methods, FedEP does utilize the variational approximation to obtain the global posterior. However, determining whether MCMC or VA is more suitable for federated posterior inference is in general a complex problem, as their effectiveness depends on the specific context. Due to the data heterogeneity issue, regularization at the client level is often necessary to mitigate bias in the local posteriors. Our FedMDMI employs the model-data mutual information regularization alongside the MCMC for posterior inference. In contrast, FedEP uses the posterior from the previous round to regularize the current local posterior. However, in some practical federated scenarios with low client participation rates, this retained local posterior in FedEP can become very stale, thus reducing its effectiveness in addressing the data heterogeneity issue. This also contributes to the less stable and slower convergence observed for FedEP compared to our FedMDMI.

---

> > > ### Comment · Reviewer_4nGu · 2024-08-13
> > > **Reply**
> > >
> > > Thanks for the additional explanations and clarifications. Your responses clarify most of my questions and confusion. Please include this additional information about the time and memory complexity in the supplementary. Also please add the variances and error bars to your tables and plots. In the discussion section of your paper, elaborating on MCMC vs. variational methods would be very helpful to place your work in the broader literature.
> > >
> > > I'm going to increase my rating of the paper. Although there are clear and significant improvements in the performance and calibration results compared to competing methods, the improvements look marginal. Nevertheless, I think the paper deserves acceptance as it provides thorough experimental and theoretical arguments and results show clear advantages to other models (in terms of performance and runtime).

---

> > > > ### Author Response · Authors · 2024-08-13
> > > > **Official Comment by Authors:**
> > > >
> > > > Dear Reviewer 4nGu,
> > > >
> > > > We would like to express our gratitude once again for the time and effort you have dedicated in reviewing our paper, including the rebuttal and discussion phases. Your insightful comments are invaluable for enhancing our work. In the final version of our manuscript and supplementary material, we will include these additional experimental results, complexity analysis regarding the time and memory, comparison of MCMC and variational inference, and the variances and error bars in the tables and plots.

---

### Author Rebuttal · Authors · 2024-08-07

Thanks for the comments. Below is our response to common cocerns, with new tables in the attached PDF.

**Comment 1-Complexity analysis :** We provide complexity analysis w.r.t. time, storage, and communication of different methods. We further empirically examine the relationship between computation time, data heterogeneity, and model dimension within a communication round.

1) We begin by defining the dimensions of neural network as $d$. At each round, we adopt SGLD (Stochastic Gradient Langevin Dynamics) to estimate the local posterior. Specifically, compared to SGD employed in **FedAvg** and **FedBE**, **our FedMDMI** entails an additional step of generating Gaussian noise with $d$ dimensions, which is subsequently incorporated into each model update iteration. This causes additional $\mathcal{O}(d)$ time and $\mathcal{O}(d)$ memory. In contrast, at each client, **FedPA** uses dynamic programming to approximate the inverse matrix $d \times d$ of neural network, incurring additional $\mathcal{O}(l^2d)$ time and $\mathcal{O}(ld)$ memory, where $l$ is number of posterior samples. Similarly, **FedEP** also requires approximating the covariance as the inverse Hessian, introducing additional $\mathcal{O}(d^3)$ time and $\mathcal{O}(d^2)$ memory.

For communication and aggregation at server, our **FedMDMI**, along with **FedAvg** and **FedPA**, requires $\mathcal{O}(md)$ time and $\mathcal{O}(md)$ memory, where $m$ denotes number of clients. In contrast, **FedEP** requires $\mathcal{O}(md)$ time and $\mathcal{O}(md^2)$ memory. **FedBE** not only requires $\mathcal{O}((m+1+l)d)$ time and $\mathcal{O}((m+1+l)d)$ memory, where $l$ denotes number of global model samples, but also performs knowledge distillation at server using unlabeled data, which is both memory- and time-intensive.

2) We also conduct experiments to evaluate running time of each communication round for these algorithms. Taking CIFAR100 dataset with participation rate $\frac{p}{m}=0.1$ as example, the average time to execute a communication round for LeNet and ResNet-18 with GEFORCE GTX 1080 Ti is:

**LeNet**:
FedAvg: 5.29 s
FedPA: 6.94 s
FedEP: 11.25 s
FedBE: 17.36 s
FedMDMI: 5.36 s

**ResNet-18**:
FedAvg: 13.05 s
FedPA: 16.87 s
FedEP: 25.10 s
FedBE: 43.06 s
FedMDMI: 13.88 s

This shows that time consumed by our FedMDMI per communication round does not exhibit significant difference than FedAvg. The time required by FedBE is much larger than other methods. As network size increases, the time taken by all baselines also increases, aligning with our intuition.

Finally, data heterogeneity does not affect local computation time when numbers of local samples and updates are fixed.

**Comment 2-Additional Baeslines:** We will include these references in the revised manuscript. The survey provides an overview of Bayesian federated learning (BFL). For example, FedHB proposes a hierarchical BFL approach, using hierarchical Bayesian modeling to describe the generative process of clients' local data with local models governed by a higher-level global variate. FedSI proposes a personalized BFL, performing posterior inference over an individual subset of network parameters for each client, while keeping other parameters deterministic.

Inspired by FedRep and FedSI, we apply our algorithm to personalized Bayesian FL, named PerFed-MDMI. FedRep learns a shared data representation across clients and unique local heads for each client to fulfill their personal objectives. FedSI further updates the distributions of model parameters over the representation layers and sends these updated distributions to the server for global aggregation in training. Model parameters of the decision layers are then fine-tuned during evaluation phase. For subnetwork (i.e., representation layers) inference, instead of using Linearized Laplace Approximation from FedSI, PerFed-MDMI employs our model-data mutual information regularization. Also, since our posterior inference method involves no covariance matrix estimation, we do not consider selecting a smaller subnetwork to reduce computation and storage overhead.

Empirically, we first evaluate our method against FedProx and FedHB on traditional FL task, named global prediction. Table 9 shows that our method outperforms FedProx and FedHB.

For personalized FL task, we compare our newly designed method, PerFed-MDMI, with FedAvg and FedRep, presenting improvement in Table 10, where all data at each client is split into 70\% training set and 30\% test set. This indicates that our MI regularization is compatible with FedRep and FedSI, effectively promoting personalized FL. Additionally, since our focus is on training a global posterior rather than the personalized posteriors, we did not directly compare the accuracy with FedSI, FedPop, and other baselines in personalized FL due to the time constraint.

**Comment 3-Comparison of MCMC and Variational Approximation (VA):**

We use SGLD instead of VA for three reasons.

1) Optimal posterior obtained in our FedMDMI is the Gibbs posterior, and SGLD has been proven efficient and effective in large-scale Gibbs posterior inference.

2) As shown in Line 212, there has also been pioneering work on obtaining a Gibbs posterior through VA. But this method results in at least doubling the communication overhead due to transmission of both the mean and covariance matrices. Another primary drawback of VA is its tendency to yield biased posterior estimates for complex posterior distributions. This issue is further compounded in FL, where data heterogeneity across clients already contributes to biased local posteriors, and this bias may be exacerbated by VA.

3) As shown in Figs. 3(c)(d) and 4(d), convergence rate of our FedMDMI may not be the fastest, especially when compared to the optimization-based method, SCAFFOLD. But it still slightly outperforms Bayesian-based baselines in most cases, showing that SGLD in FL is efficient and does not significantly reduce the convergence rate.

---

### Decision · Program_Chairs · 2024-09-25

**Decision:**

Accept (poster)

**Comment:**

The paper addresses a timely and important problem in federated learning by proposing a novel approach to tackle data heterogeneity and scarcity using Bayesian modeling for uncertainty estimation and mutual information regularization. The proposed methodology is well-motivated and combines theoretical arguments with empirical results, making it a strong contribution to the field. The clear and logical writing, along with the detailed theoretical derivations, further enhances the paper's strength. Notably, the paper provides a thorough exploration of an underexplored area in federated learning, offering sound justifications for most design decisions and demonstrating the potential of information-theoretical modeling in this context.

Despite its strengths, the paper has some limitations that reviewers noted. The performance gains observed were marginal, and there were concerns about the lack of comprehensive comparisons with recent state-of-the-art baselines specifically designed to address data heterogeneity in federated learning. Some important baselines were missing from the experiments, and the empirical evaluations were conducted on small-scale models, raising questions about the method's scalability to more complex architectures.

Overall, while there are areas for improvement, the paper's strengths in tackling a crucial problem with novel and theoretically grounded methods justify its acceptance.